# Cancer immune therapy with PD-1-dependent CD137 co-stimulation provides localized tumour killing without systemic toxicity

Yunqian Qiao[1,2], Yangmin Qiu [1,2], Jie Ding[1], Nana Luo[1], Hao Wang[1], Xiaomin Ling[1], Jiya Sun[1], Zhihai Wu[1], Yisen Wang[1], Yanpeng Liu[1], Feifei Guo[1], Ta Sun[1], Wanwan Shen[1], Min Zhang[1], Dongdong Wu[1], Bingliang Chen[1], Wei Xu [1✉] & Xuan Wang [1✉]

Expression of the cell surface receptor CD137 has been shown to enhance anti-cancer T cell function via engagement with its natural ligand 4-1BBL. CD137 ligation with engineered ligands has emerged as a cancer immunotherapy strategy, yet clinical development of agonists has been hindered by either toxicity or limited efficacy. Here we show that a CD137/PD-1 bispecific antibody, IBI319, is able to overcome these limitations by coupling CD137 activation to PD-1-crosslinking. In CT26 and MC38 syngeneic mouse tumour models, IBI319 restricts T cell co-stimulation to PD-1-rich microenvironments, such as tumours and tumour-draining lymph nodes, hence systemic (liver) toxicity arising from generalised T cell activation is reduced. Besides limiting systemic T cell co-stimulation, the anti-PD-1 arm of IBI319 also exhibits checkpoint blockade functions, with an overall result of T and NK cell infiltration into tumours. Toxicology profiling in non-human primates shows that IBI319 is a well-tolerated molecule with IgG-like pharmacokinetic properties, thus a suitable candidate for further clinical development.

[1] Innovent Biologics (Suzhou) Co. Ltd, Suzhou, China. [2] These authors contributed equally: Yunqian Qiao, Yangmin Qiu. ✉email: xu.wei@innoventbio.com; xuan.wang@innoventbio.com

C D137 (4-1BB, TNFRSF9) belongs to the tumour necrosis factor receptor superfamily (TNFRSF) and is expressed on various cell types, including T cells and natural killer (NK) cells, upon activation and constitutively at lower levels on monocytes, neutrophils, dendritic cells (DCs), and some tissue cells, such as lung and liver endothelial cells[1–3]. The physiological signal transduction mediated by CD137 is induced by its natural ligand 4-1BBL, which is a type II membrane protein in the TNF superfamily (TNFSF)[4]. Similar to other TNFSF members, 4-1BBL presents as a membrane-bound or soluble homotrimeric complex that mediates the trimerization of CD137 and subsequent recruitment of specific TNF receptor-associated factors (TRAF1, TRAF2 and TRAF3) and initiation of downstream signalling cascades, such as NFκB, ERK, JNK, and p38 MAPK pathways[5,6].

Over the past decade, increasing efforts have been made to tackle CD137 as a potential second-generation immuno-oncological target to further stimulate tumour-specific T cells[7–9]. CD137-expressing T cells were found to display a higher degree of T cell activation and less exhaustion than CD137-negative T cells within tumour-infiltrating lymphocytes (TILs) in patients with ovarian cancer. A CD137 agonist further enhanced the anti-PD-1 antibody-mediated reinvigoration of exhausted CD8 + TILs from both the primary sites and metastatic sites[10], indicating the rationale for targeting CD137 in combination with checkpoint blockade. However, the clinical trials evaluating two CD137-specific monoclonal antibodies (mAbs) were halted due to either intolerable hepatotoxicity (urelumab, BMS[11]) or low efficacy (utomilumab, Pfizer[12]). With the increasing number of clinical studies performed to evaluate immunotherapeutic agents, it is important to avoid potential immune-related adverse events (irAEs) that could be life-threatening[13,14]. Indeed, neither the mechanisms of how agonistic anti-CD137 antibodies induce receptor trimerization and downstream signalling nor the causes of hepatoxicity have been fully addressed[9,15]. Moreover, structural studies have revealed that urelumab binds to the cysteine-rich pseudo repeat 1 (CRD1) of the CD137 extracellular domain, whereas 4-1BBL and utomilumab bind to the CRD2/3 region with slightly distinguished epitopes[15], suggesting a close correlation between the binding epitope and activation efficacy. Nevertheless, reducing off-target toxicity while retaining antitumour efficacy is a continuing challenge in advancing CD137 agonists into clinical applications, and overcoming this issue will likely require consideration of the Fc function, affinity and binding epitope properties of the desired new molecule.

T cell activation requires antigen recognition by the T cell receptor (TCR, signal 1), MHC-independent co-stimulatory signalling, including signalling via CD137 (signal 2), and cytokine priming (signal 3)[16]. Since the binding of programmed cell death 1 (PD-1) to its ligands programmed death-ligand 1 and 2 (PD-L1/PD-L2) provides a negative feedback signal to counteract TCR activation via the protein tyrosine phosphatase SHP-2[17], PD-1/PD-L1 blockade mainly affects the outcome of signal 1[18]. Therefore, simultaneously blocking PD-1/PD-L1, while activating CD137 has the potential to generate a synergistic effect on T cell activation that leads to better antitumour activity via signals 1 and 2.

Here we show a PD-1/CD137 bispecific antibody, IBI319. The anti-CD137 arm of IBI319 has a binding epitope similar to that of natural 4-1BBL but a significantly lower binding potency than that of the anti-PD-1 arm. This design ensures a preferential distribution of the antibodies to PD-1-expressing cells, namely, T cells and NK cells infiltrating the tumour and/or in tumour-draining lymph nodes (TDLNs) and avoids the liver toxicity caused by the systemic distribution of CD137 agonists. Despite distribution, binding to the receptor PD-1 crosslinks the molecules, leading to CD137 trimerization, and downstream signalling. IBI319 has potential in enhancing the clinical response of immune checkpoint blockades while reducing toxicity caused by conventional CD137 agonists.

## Results

**IBI319 activates CD137 in a PD-1 dependent manner**. IBI319 was designed as a fully humanised IgG1 molecule comprising a bivalent Fab fragment that bound to PD-1 and CD137 separately (Fig. 1a). The Fab region of the anti-PD-1 arm was from sintilimab[19,20] (αPD-1), an approved PD-1 blocker, with minor modifications, such as M54S in the heavy chain, to eliminate methionine oxidation. The anti-CD137 arm was modified from a previously generated anti-CD137 antibody (αCD137 mAb) with the mutations M101A and G106A in the CDR region of the heavy chain to eliminate methionine oxidation and reduce binding affinity (patents WO2020018354A1 and CN202010055492.X). This engineering allowed IBI319 to maintain a high affinity for PD-1 (dissociation constants ($K_D$s): 0.10 nM via surface plasmon resonance (SPR) and 0.632 nM via biolayer interferometry (BLI)) and a low affinity for CD137 ($K_D$s: 394 nM via SPR and 719 nM via BLI) (Fig. 1a, b; Supplementary Fig. 1a) to achieve strong blocking of PD-1 and appropriate agonism of CD137. A ligand-blocking assay revealed that IBI319 could completely block the binding of CD137 ligand (CD137L) to CD137 (Fig. 1c), indicating the antibody binding epitope overlapped with the binding site of the natural ligand. Indeed, IBI319 bound to CD137 at an affinity similar to that of CD137L; monovalent CD137L binds to CD137 with a $K_D$ of 680 nM using SPR (Fig. 1a)[15]. L234A, L235A, and N297Q mutations were introduced in the Fc region of IBI319 to eliminate binding to FcγRs (Supplementary Fig. 1b) and Fc-mediated ADCC effector function (Supplementary Fig. 1c). The binding of IBI319 to FcRn was not affected (Supplementary Fig. 1b). For control purposes, the FcγR binding function of the αCD137 mAb was also silenced, and the αPD-1 mAb was in an IgG4 form (Supplementary Fig. 1b).

Since primary human CD3 + T cells activated by anti-CD3/CD28 antibody-coated beads expressed CD137 and PD-1 (Fig. 1d, Supplementary Fig. 1d), reminiscent of TILs in tumour-bearing mice (Supplementary Fig. 3f), we evaluated the binding of IBI319 to primary T cells. IBI319 bound to CD4 + and CD8 + T cells at 50% maximal effect ($EC_{50}$) values of $1.39 \pm 0.45$ nM and $0.90 \pm 0.19$ nM, respectively. However, almost no specific binding of the αCD137 mAb to activated T cells was detected (Fig. 1e). We assumed that the low binding of the αCD137 mAb to primary T cells was due to its low affinity, the lower CD137 expression level on T cells than on the cell line tested (Supplementary Fig. 1d) and possibly the bivalent mAb without crosslinking lacking the capability to stabilise the binding with CD137, as the CD137L-CD137 complex was reported to ensure the recruitment of downstream TRAF proteins that could further stabilise the binding interaction[21].

To dissect the cell-surface binding affinity of IBI319 for CD137, a Jurkat cell line overexpressing CD137 (Jurkat-CD137-NFκB-Luc, Jurkat-CD137 for short) was used to provide CD137, and a CHO-S cell line overexpressing PD-1 was generated to provide a PD-1 handle (CHO-S-PD-1). Indeed, only in the presence of CHO-S-PD-1 cells (Jurkat-CD137 cells co-cultured with CHO-S-PD-1 cells) did IBI319 show high binding (flow cytometry (FACS) mean fluorescence intensity (MFI) value) to Jurkat-CD137 cells, whereas the αCD137 mAb displayed a relatively low binding affinity regardless of the presence of CHO-S-PD-1 cells (Fig. 1f). Moreover, the bridging of these two cell lines was detected by identifying the corresponding double-positive population via FACS and was dependent on the IBI319 concentration (Supplementary Fig. 1e, f). These results indicate

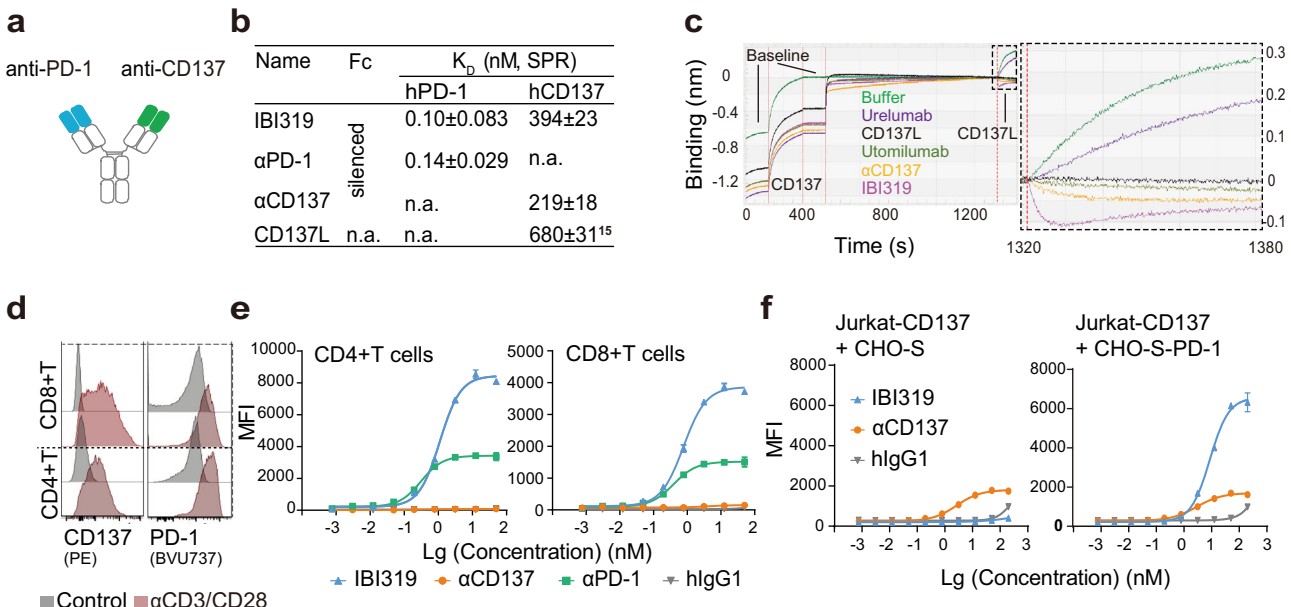

**Fig. 1 IBI319 activates CD137 in a PD-1 dependent manner. a** Structure diagram of IBI319 on a human IgG1 backbone. IBI319 consists of a Fab region binding to PD-1 and CD137 and an Fc fragment with null effector function. **b** The dissociation constants ($K_{D}$s) of IBI319, αPD-1, and αCD137 for human PD-1 and human CD137, respectively, determined via SPR (3 independent experiments; mean ± standard deviation (SD); n.a. not analysed). **c** CD137 ligand-blocking properties of IBI319, αCD137, in-house-generated urelumab and utomilumab via BLI. **d** PD-1 and CD137 expression on CD4 + and CD8 + T cells from a representative healthy donor before (black) and after activation by anti-CD3/CD28 antibody-coated beads (red) measured by flow cytometry (FACS). **e** Binding affinities of IBI319 and control antibodies for activated CD4 + T cells and CD8 + T cells measured by FACS. **f** Binding affinities of IBI319 and control antibodies for Jurkat-CD137 cells co-cultured with CHO-S or CHO-S-PD-1 cells determined by FACS and calculated from the CTV + CFSE + population. **e**, **f** two technical replicates of one representative experiment from three independent experiments; mean and SD.

that IBI319 engages both targets simultaneously and that its anti-CD137 arm relies on the crosslinking provided by the anti-PD-1 arm binding to PD-1.

**IBI319 blocks the PD-1/PD-L1 interaction and selectively enhances CD137 activity in a PD-1-dependent manner.** We next determined the functionality of IBI319. A PD-1/PD-L1 blocking bioassay system consisting of a Jurkat cell line expressing human PD-1 and an NFAT-driven luciferase reporter gene (Jurkat-PD-1-NFAT-Luc), and a CHO-K1 cell line expressing human PD-L1 and an artificial cell-surface TCR activator (CHOK1-PD-L1-TCRa) were utilised to measure the potency of the IBI319-PD-1 arm. Weaker PD-1/PD-L1 blocking activity was observed for IBI319 ($EC_{50}$ 8.08 ± 1.51 nM) than for the αPD-1 mAb ($EC_{50}$ 1.09 ± 0.24 nM), which could be due to the monovalent binding site of IBI319 for PD-1 (Fig. 2a). To specifically evaluate antibodies triggering CD137 signalling, the Jurkat-CD137 cells was co-cultured with antibodies with or without CHO-S-PD1 cells. Since the luciferase activity in Jurkat-CD137 cells was regulated by NFκB, its induction reflected only CD137 function but not the influence of PD-1 blockade. As expected, only upon the addition of CHO-S-PD-1 cells could IBI319 strongly stimulate luciferase activity (Fig. 2b-left and -middle), and this activity was dependent on the number of PD-1 molecules represented by the ratio of CHO-S-PD-1 cells to Jurkat-CD137 cells (Fig. 2b-right). In contrast, luciferase activity was only weakly induced by the αCD137 mAb or a combination of the αCD137 and αPD-1 mAbs at high concentrations (Fig. 2b). We assumed that the weak luciferase activity induced by the αPD-1 mAb in Jurkat-CD137 cells was a nonspecific background signal. These results demonstrate intercellular (*trans*) activation and the requirement for PD-1-expressing cells to mediate the CD137 agonistic activity of IBI319.

To further examine whether IBI319 can bind to both the PD-1 and CD137 and activate CD137 on a single cell (*cis*-interaction), we generated a Jurkat-based reporter cell line expressing CD137, NFκB-Luciferase and PD-1 (Jurkat-CD137-NFκB-Luc2P-PD-1). Cell-surface antigen quantification revealed that Jurkat-CD137-NFκB-Luc2P-PD-1 cells expressed 9924 CD137 molecules and 21422 PD-1 molecules per cell on average, which were within the same ranges as those of primary T cells derived from CD3/CD28 bead-stimulated peripheral blood mononuclear cells (PBMCs) (14644 CD137 molecules and 16038 PD-1 molecules per cell, Supplementary Fig. 2a). Jurkat-CD137-NFκB-Luc2P-PD-1 cells expressed similar levels of CD137 as the Jurkat-CD137 cell line (11246 CD137 molecules per cell) but lower levels of PD-1 than the CHO-S-PD-1 cell line used in previous experiments (262026 PD-1 molecules per cell) (Supplementary Fig. 2a). Jurkat-CD137 and Jurkat-CD137-NFκB-Luc2P-PD-1 cells displayed similar NFκB activation potentials after TNFα stimulation (Supplementary Fig. 2b). To exclude a crosslinking signal transduced via *trans*-cell interactions between the neighbouring cells in 2D culture (Fig. 2c-left), we introduced a 3-dimensional (3D) culture system (FCeM® polymer system) to obtain a single-cell suspension and thus avoid cell-cell interactions. The 3D culture system significantly reduced the IBI319-induced luciferase signal of the parental reporter cell line induced through *trans*-cell interactions (Jurkat-CD137 cells co-cultured with CHO-S-PD-1 cells), whereas Jurkat-CD137-NFκB-Luc2P-PD-1 cells retained approximately 46% of the luciferase signal observed in 2D culture (Fig. 2c-right), indicating the contribution of the *cis*-cell interaction-transduced signal.

In addition to the reporter systems, a mixed leukocyte reaction (MLR) assay using primary human CD3 + T cells and monocyte-derived mature DCs was performed to test the capacity of IBI319 to enhance an allogeneic T cell response. T cells in the allogeneic incubation system upregulated PD-1 and CD137, enabling the

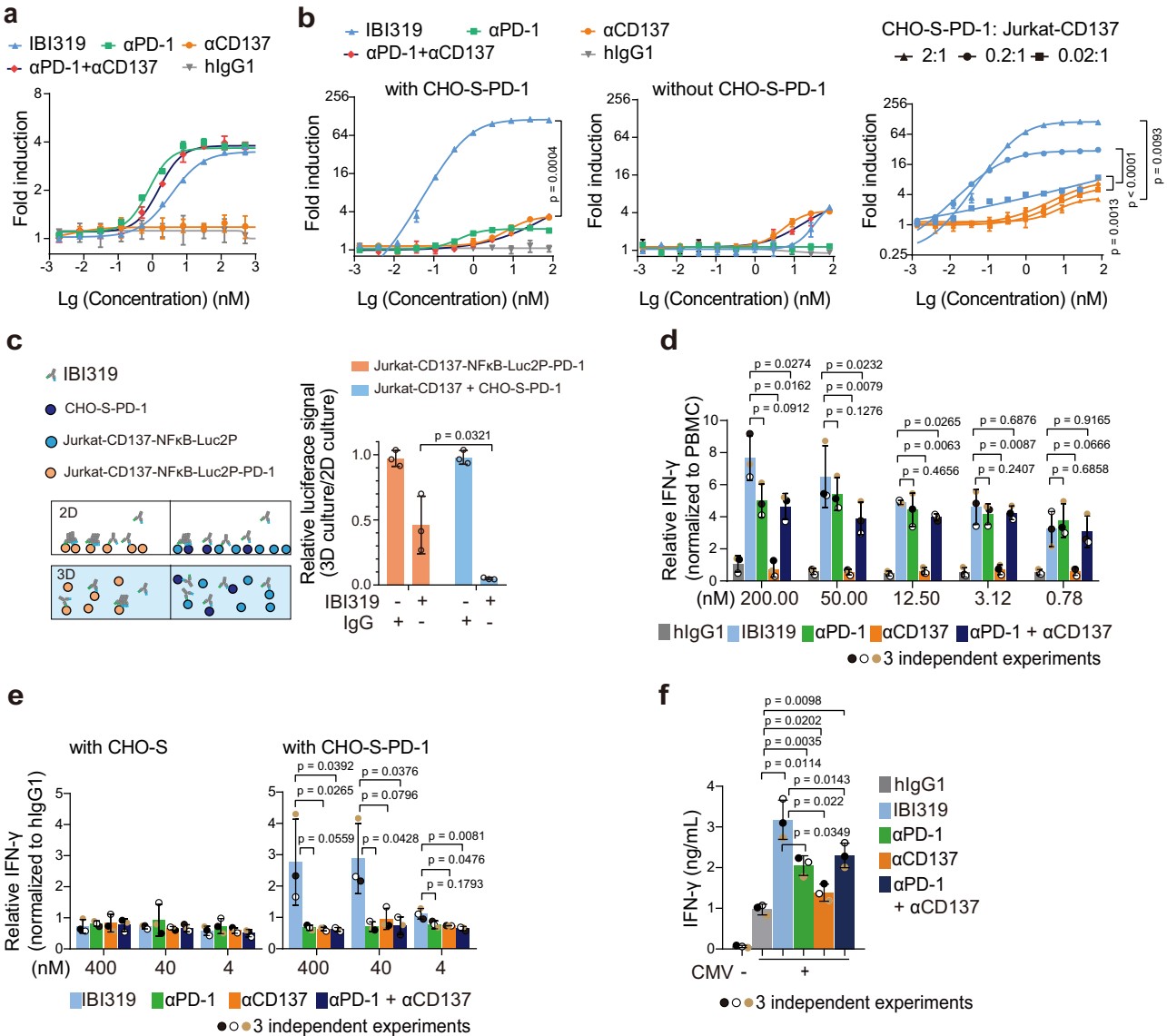

**Fig. 2 IBI319 blocks the PD-1/PD-L1 interaction and selectively enhances CD137 activity in a PD-1-dependent manner. a** PD-1/PD-L1 blocking activities of different antibodies presented as NFAT-mediated luciferase activity. Fold induction was calculated as the mean relative light units (RLU) of tested antibodies/mean RLU of the no antibody control. **b** CD137 agonist activities of different antibodies presented as NFκB-mediated luciferase activity. Jurkat-CD137-NFκB-Luc (Jurkat-CD137) cells were co-cultured with two times the number of CHO-S-PD-1 cells (left), without CHO-S-PD-1 cells (middle), and with different amounts of CHO-S-PD-1 cells (right). Fold induction was calculated as the mean RLU of tested antibodies/mean RLU of the no antibody control. **c** Three dimensional (3D) and 2 dimensional (2D) cell culture systems to test the *cis*- and *trans*-cell crosslinking of IBI319. (Left) The 3D and 2D cell culture models when treated with IBI319. (Right) The relative luciferase signals of the Jurkat-CD137-NFκB-Luc2P-PD-1 alone and Jurkat-CD137 + CHO-S-PD-1 groups were calculated with the equation: (mean RLU of tested antibodies in 3D culture/mean RLU of the control in 3D culture)/(mean RLU of tested antibodies in 2D culture/mean RLU of the no antibody control in 2D culture). **d** Relative IFN-γ production in an MLR assay. Antibodies were 4-fold serially diluted from 200 nM. **e** Relative IFN-γ production level in a T cell co-stimulation assay. Primary T cells were treated with 1 μg/mL CD3/CD28 activator, and antibodies were serially diluted 10-fold from 400 nM in the presence of CHO-S or CHO-S-PD-1 cells. **f** IFN-γ production level in a cytomegalovirus (CMV) T cell memory recall assay. PBMCs from a donor previously infected with CMV were co-cultured with a CMV peptide pool and antibodies at 50 nM for 4 days (data for three donors in Supplementary Fig. 2e). **a** and **b** show two technical replicates of one representative experiment from three independent experiments. mean and SD. **c, d, e, f** show individual values of three independent experiments and the mean ± SD. Statistics were performed in the indicated groups: two-way ANOVA with Tukey's multiple comparisons test (**b**) and two-sided paired *t*-test (**c, d, e, f**); *p* values are indicated in the graphs.

analysis of functions related to PD-1 blockade and CD137 agonism (Supplementary Fig. 2c- bottom). In comparison to αPD-1, αCD137 mAbs and the combination of the αPD-1 and αCD137 antibodies, IBI319 induced the highest IFN-γ production at high concentrations after 5 days of T cell and DC co-incubation (Fig. 2d, absolute values of individual donors in Supplementary Fig. 2c-top). In particular, IBI319 showed a significantly stronger effect than the αPD-1 + αCD137

combination treatment, in which the αPD-1 and αCD137 antibodies were added at equivalent concentrations as IBI319, suggesting that the contribution of the anti-CD137 arm in IBI319 is due to the anti-PD-1 arm in the same molecule (Fig. 2d, absolute values of individual donors in Supplementary Fig. 2c). Similarly, when primary T cells were activated by a small amount of CD3/CD28 T cell activator (1 μL/mL), in which PD-1 expression was sub-optimally upregulated (Supplementary

Fig. 2d-bottom), IBI319 further enhanced IFN-γ production at 400 and 40 nM when CHO-S-PD-1 cells were included in the culture, whereas only minimal activation occurred in other treatment groups (Fig. 2e, absolute values of individual donors in Supplementary Fig. 2d-top). Since antigen-specific T cells contribute to the recognition of tumour antigens and may upregulate PD-1 and CD137 in the tumour site[22], these cells could serve as a more physiologically relevant model. Therefore, we evaluated the co-stimulatory function of IBI319 in a cytomegalovirus (CMV) memory recall assay using PBMCs from three donors previously infected with CMV, as T cells specific for CMV antigens comprise a fraction of the memory T cell pool. The CMV peptide pool could induce IFN-γ production, and IBI319 further enhanced this effect better than αPD-1 or the combination of αPD-1 and αCD137, indicating the contributions of both arms of IBI319 (Fig. 2f, individual donors in Supplementary Fig. 2e). Finally, since the anti-CD137 arm of IBI319 has a relatively low affinity compared to other CD137 binders that are in clinical development, we investigated whether using anti-CD137 antibody sequences with higher affinities in the bispecific antibody could achieve better agonism. We constructed PD-1/CD137 bispecific antibodies using anti-CD137 sequences from urelumab (PD-1/Ure) and utomilumab (PD-1/Uto). PD-1/Ure and PD-1/Uto bound to CD137 with $K_{D}$s of 19.2 and 49.3 nM, respectively (Supplementary Fig. 2f). In a Jurkat-CD137-luc reporter assay, all 3 bispecific antibodies showed PD-1-dependent CD137 activity. IBI319 was as efficacious as PD-1/Uto, and PD-1/Ure showed the highest activity among all tested bispecific molecules (Supplementary Fig. 2g). Interestingly, IBI319 showed better primary T cell co-stimulation than PD-1/Ure and PD-1/Uto (Supplementary Fig. 2h). The inconsistency in the link between the efficacy and affinity remains to be resolved in the future but could be due to differences in the binding epitope, which would influence the crosslinking of two targets/cells. Therefore, the optimal CD137-specific sequence in a bispecific molecule requires a good combination of various factors, including affinity, the binding epitope and probably the molecular format.

Taken together, these data indicate that IBI319 has better in vitro activity for enhancing T cell activity than αPD-1 and αCD137 mAbs or their combination and functions in a *cis*- and *trans*-cell PD-1-dependent manner.

**IBI319 shows strong antitumour efficacy while avoiding hepatotoxicity in vivo**. We then sought to explore the in vivo efficacy and safety profile of IBI319. Previous studies showed that the combination of an anti-CD137 antibody with an anti-PD-L1 or anti-PD-1 antibody achieved better efficacy than monotherapy treatments in an HCC827 tumour model established with NSG mice implanted with human T cells or in a mouse UN-SCC680AJ syngeneic model[23,24], suggesting a synergistic effect was achieved by combining checkpoint blockade and CD137 stimulation. Therefore, in addition to confirming that IBI319 retains this synergy, we focused on addressing the potential liver toxicity that can be induced by targeting CD137. Since the function of the anti-CD137 arm strongly relies on the presence of proximal PD-1 molecules, we chose PD-1 blockade-responsive tumour types, i.e., the moderately responsive CT26 model and the responsive MC38 model[25,26]. To maintain the affinity of IBI319 to human antigens, two target-humanised transgenic mouse models were established: a BALB/c model in which the extracellular domains of PD-1, CD137 and PD-L1 were replaced by those of the corresponding human proteins (BALB/c-hPD-1/hCD137/hPD-L1) and a similar C57BL/6 model harbouring the hPD-1 and hCD137 extracellular domains (C57BL/6-hPD-1/hCD137). Of note, it was previously reported that murine PD-L1 (mPD-L1) binds to human PD-1

(hPD-1) with an affinity similar to that of hPD-L1 binding to hPD-1[27,28] (Supplementary Fig. 3a, b), suggesting that the hPD-1 expressed by transgenic mice can bind to mPD-L1 expressed by MC38 cells. In BALB/c-hPD-1/hPD-L1/hCD137 mice implanted with the murine CT26 colorectal cancer cell line overexpressing human PD-L1 (CT26-hPD-L1), 7 days after the third antibody injection, IBI319 showed an enhanced efficacy compared to the αPD-1 mAb and αCD137 mAb (undetectable efficacy) (Fig. 3a) and was as efficacious as the αPD-1 mAb + αCD137 mAb combination treatment (Supplementary Fig. 3c in a separate experiment), indicating the contributions from both the anti-PD-1 and anti-CD137 arms of IBI319. In C57BL/6-hPD-1/hCD137 mice bearing another colorectal tumour cell line (MC38), compared to hIgG, IBI319 showed clear dose-dependent antitumour efficacy at 0.3, 1, 3, and 10 mg/kg with tumour growth inhibition (TGI) rates of 3, 42, 33 and 91%, respectively (Fig. 3b-left). Three out of 7 mice treated with 10 mg/kg IBI319 showed complete tumour eradication (Supplementary Fig. 3e). Similar to the CT26 model, the MC38 model showed that combined treatment with the αPD-1 and αCD137 mAbs at 0.5 + 0.5 or 5 + 5 mg/kg was as efficacious as IBI319 at equivalent doses (1 and 10 mg/kg, respectively) (Fig. 3b-right). This synergistic effect is similar to effects identified in previous reports[23,24]. Consistent with the efficacy of IBI319, both the PD-1 and CD137 were upregulated on TILs in MC38 tumours compared to peripheral T cells, as measured by FACS (Supplementary Fig. 3f). Of note, more MC38 tumour-derived TILs expressed CD137 on day 28 after tumour inoculation than on day 11 after tumour inoculation (Supplementary Fig. 3f), suggesting that CD137 on TILs was upregulated with tumour progression. A single-cell RNA-sequencing (scRNA-seq) analysis of CD45 + TILs in MC38 tumours revealed that both the two classes of T cells (class I and class II) and NK cells were increased in the IBI319 and αPD-1 + αCD137 mAb combination treatment groups in comparison to the αPD-1 treatment group, suggesting that the contribution of CD137 stimulation to efficacy is mainly mediated through T and NK cells (Fig. 3c, d; Supplementary Fig. 4). The hIgG1 and αCD137 mAb groups were not analysed, as the tumour sizes were much larger than those of the IBI319 (3 mg/kg), αPD-1 (3 mg/kg) and αPD-1 + αCD137 (1.5 + 1.5 mg/kg) groups, and the TIL numbers were not comparable.

Weak CD137 binders were previously reported to achieve efficacy via crosslinking of the anti-CD137 antibody through the Fc region binding to FcRs[29]. Therefore, to further specifically dissect the efficacy of the anti-CD137 arm of IBI319 in vivo, we constructed a control anti-CD137 mAb in an hIgG1 form that bound to mouse Fc receptors (Fig. 3e). For a fair comparison in terms of dose, anti-CD137 and anti-PD-1 mAbs were generated as monovalent antibodies with the other arm binding to hen egg white lysozyme (Hel) (CD137/Hel-Fc, CD137/Hel-Fc-null, and PD-1/Hel). The anti-Hel arm did not recognise any human or mouse antigens and was used to maintain the IgG structure. The active Fc function of CD137/Hel-Fc did not lead to T cell depletion due to ADCC (Supplementary Fig. 3g-right). Consistent with previously reported weak CD137 binders, CD137/Hel-Fc became more efficacious than CD137/Hel-Fc-null (TGI rates: 63.1% and 30.2%, respectively) at 10 mg/kg (Supplementary Fig. 3g-left). At 3 mg/kg, IBI319 showed stronger efficacy than PD-1/Hel or CD137/Hel-Fc alone, again indicating the synergistic effect of the two arms. The efficacy achieved with combination therapy with CD137/Hel-Fc and PD-1/Hel in this experiment was stronger than that achieved with IBI319, presumably because CD137/Hel-Fc functions in both the Fc-dependent and Fc-independent ways when combined with an anti-PD-1 antibody (Fig. 3e).

Although efficacious, IBI319 did not induce immune cell infiltration to the liver after three doses (Q.W.) in

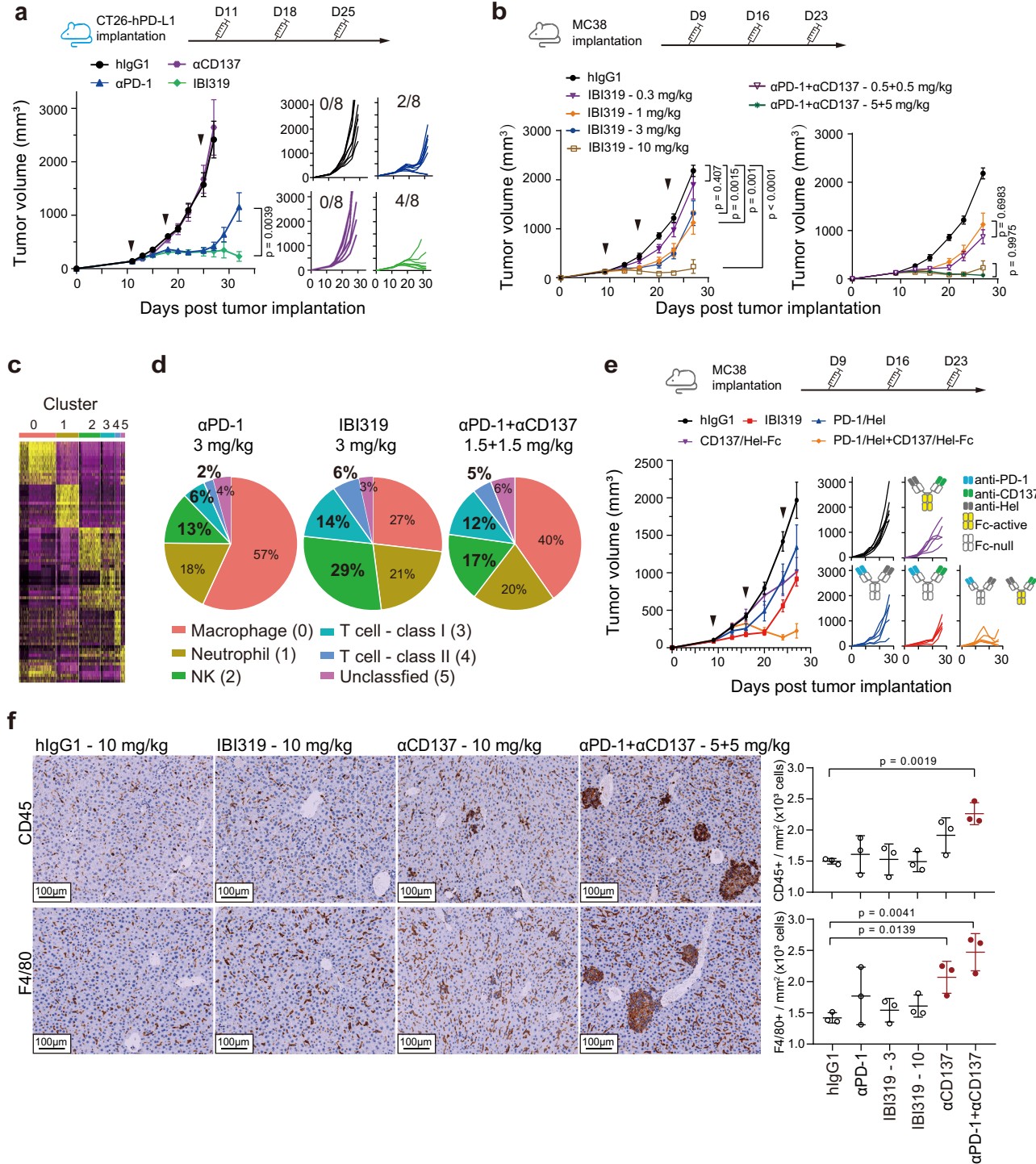

**Fig. 3 IBI319 shows strong antitumour efficacy while avoiding hepatotoxicity in mouse models. a** (Left) Tumour volumes of CT26-hPD-L1 tumours in BALB/c-hPD-1/hPD-L1/hCD137 mice treated with the indicated antibodies (20 mg/kg, i.p.) on days 11, 18 and 25 post-tumour implantation ($n = 8$ per group). The mice in the hIgG1 and αCD137 groups were euthanized on day 27 post-tumour implantation due to ethical issues. (Right) Spaghetti plots show the individual tumour volumes in each group. The numbers in the plots indicate tumour-free mice/all. **b** Tumour volumes of MC38 tumours in C57BL/6-hPD-1/hCD137 mice treated with the indicated antibodies on days 9, 16 and 23 ($n = 7$ per group). **c** and **d** Single-cell RNA sequencing of CD45 + cells in MC38 tumours after antibody treatments. **c** A thumbnail heatmap showing the expression of signature genes in each cell cluster (detailed heatmap in Supplementary Fig. 4b). **d** The percentages of different cell clusters in the αPD-1, IBI319 and αPD-1 + αCD137 groups. **e** Tumour volumes of MC38 tumours in C57BL/6-hPD-1/hCD137 mice treated with the indicated antibodies ($3 + 3$ mg/kg for the αPD-1/Hel + αCD137/Hel-Fc group and 3 mg/kg for all other groups) on days 9, 16 and 23 ($n = 6$ per group for the IgG1 and αPD-1/Hel + αCD137/Hel-Fc groups; $n = 5$ per group for all other groups). Hel: hen egg white lysozyme. **f** Livers from the mice in **b** were stained for CD45 and F4/80 via IHC. (Left) Representative images. (Right) Quantification of whole pathological digital slides via HALO software ($n = 3$ per group, mean and SD); images of all quantified groups can be found in Supplementary Fig. 5b. **a**, **b** and **e** Tumour volumes are presented as the mean, and the error bar represents the SEM. Statistics were performed in the indicated groups: two-way ANOVA with Tukey's multiple comparisons test (**a** and **b**) and two-sided, un-paired *t* test (**f**); *p* values are indicated in the graphs.

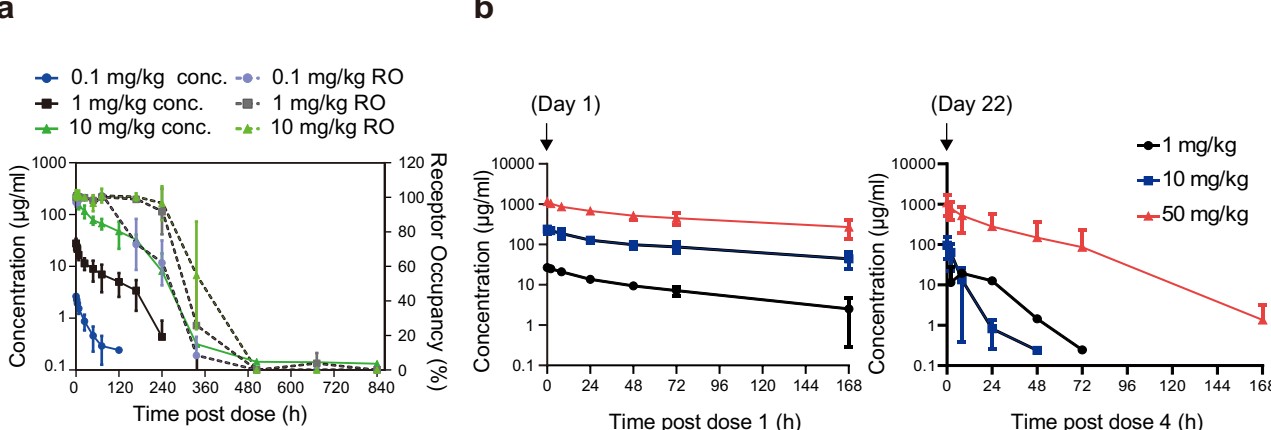

**Fig. 4 IBI319 is a tolerated molecule with an IgG-like pharmacokinetic (PK) profile suitable for further clinical development. a** Serum concentrations and receptor occupancies of IBI319 in cynomolgus monkeys following a single intravenous administration of 0.1 mg/kg, 1 mg/kg, or 10 mg/kg. $n = 6$ per group (three males and three females). conc. concentration, RO receptor occupancy, mean and SD. **b** Time-concentration curves of IBI319 following the first (day 1) and fourth (day 22) intravenous administrations of 1 mg/kg, 10 mg/kg, or 50 mg/kg IBI319 into cynomolgus monkeys. $n = 10$ per group (five males and five females); mean and SD.

C57BL/6-hPD-1/hCD137 mice, whereas the combination of αPD-1 and αCD137 mAbs resulted in an immune cell infiltrate in the liver (Supplementary Fig. 5a) that was positive for CD45 and F4/80 staining by immunohistochemistry (IHC) (Fig. 3f and Supplementary Fig. 5b), indicative of Kupffer cells/ macrophages. F4/80 was also stained in the livers of CD137/ Hel-Fc-treated C57BL/6-hPD-1/hCD137, but only a mild symptom was observed in one mouse of the CD137/Hel-Fc and PD-1/Hel combination group at 3 mg/kg each, presumably because antibody doses were low in this experiment (Supplementary Fig. 5e). Furthermore, there were no significant increases in the alanine aminotransferase (ALT) or aspartate transaminase (AST) levels or the ratio of ALT/AST[30] in the blood with IBI319 treatment or the αPD-1 and αCD137 mAb combination therapy (Supplementary Fig. 5c). The ALT/ AST was slightly increased in the CD137/Hel-Fc and PD-1/ Hel combination group at 3 mg/kg each (Supplementary Fig. 5d). Neither mouse model showed obvious body weight changes with any of the treatments (Supplementary Fig. 3h–l).

Taken together, these results suggest that IBI319 has synergistic antitumour efficacy, similar to the combination of a CD137 agonist and a PD-1 antagonist, but with undetectable immune cell infiltration into the liver in target-humanised transgenic mouse models. This change is presumably due to the preferential binding of the anti-PD-1 arm to PD-1, which is expressed in the tumour microenvironment and thus provides a sink to restrain IBI319 activity locally and avoid IBI319 systematic circulation, which may cause CD137 expression and activation in the liver.

**IBI319 is a tolerated molecule with an IgG-like pharmacokinetic (PK) profile suitable for further clinical development.** The PK and safety profiles of IBI319 were then determined in cynomolgus monkeys. The IBI319 drug serum concentration was positively correlated with increasing doses as well as the dose proportional increase in $AUC_{0-t}$ within the dose range of 0.1–10 mg/kg (single doses at 0.1 mg/kg, 1 mg/kg and 10 mg/kg) (Fig. 4a, Table 1). The drug concentration dropped at 240–360 h post-dosing, and this decrease was likely related to the occurrence of anti-drug antibodies (ADAs). The occurrence of ADAs was frequently detected on day 15 (360 h) and persisted until day 36 (864 h), which was verified by the high ADA-positive rate in all treated groups (Supplementary Table 1). Furthermore, the receptor occupancy (RO) of PD-1 was analysed in the single-dose

study, and in all dose groups, PD-1 occupancy reached saturation at 5 min postdose and exhibited a correlation between binding durability and drug concentration; that is, the RO dropped significantly on day 8 postdose in the 0.1 mg/kg group and day 15 postdose in the 1 and 10 mg/kg groups (Fig. 4a). A single-dose PK study in C57BL/6 mice revealed similar IBI319 serum concentration dynamics (Supplementary Fig. 6a).

A multiple-dose toxicokinetic (TK) study was then incorporated into a 4-week repeated-dose toxicity study with weekly intravenous (i.v.) injection of IBI319 at 1, 10 and 50 mg/kg. The results showed that the $AUC_{0-t}$ increased in a dose-dependent manner, and no significant sex difference was observed. After four doses (day 22), the drug concentration was more greatly reduced after the multiple doses than that after one dose (day 1) (Fig. 4b, Table 2). No drug accumulation was observed (Table 2). In the repeated-dose toxicity study, at 50 mg/kg, one animal was found to be moribund during drug administration on day 29, which was presumably related to immune-mediated hypersensitivity as the ADA titer was high. Other drug-related abnormalities, such as moderate mononuclear cell infiltration in multiple organs in the 50 mg/kg group and transient elevations in IL-6/C-reactive protein levels during the dosing period in the 1 and 10 mg/kg groups, were observed. In particular, adverse effects, such as liver inflammation and increased blood ALT/AST levels were not found, suggesting that IBI319 has a manageable safety profile in cynomolgus monkeys (Supplementary Fig. 6b).

Overall, these data show acceptable PK profiles for IBI319 and a sufficient safety margin to support further development of IBI319 in clinical studies.

## Discussion
In this study, we evaluated a PD-1/CD137 bispecific antibody, IBI319, aiming to develop a next-generation PD-1 blockade reagent that can further enhance antitumour activity or overcome disease relapse after anti-PD-1/PD-L1 treatment with acceptable safety[7]. Following PD-1 blockade, activated T and NK cells upregulate CD137 expression; therefore, an additional αCD137 mAb can further co-stimulate the effector function of these two cell types. The simultaneous targeting of PD-1 and CD137 by the bispecific format of IBI319 showed antitumour efficacy in vivo.

The binding of FcRs by the Fc region was reported to mediate molecular crosslinking that possibly supports CD137 trimerization, thus enhancing the efficacy of weak CD137 agonists such as

**Table 1 PK parameters of IBI319 after a single intravenous injection of 0.1, 1, or 10 mg/kg into cynomolgus monkeys (mean ± SD, $n = 6$).**

| Dose (mg/kg) | $t_{1/2}$ (h) | $AUC_{0-t}$ (h·µg/mL) | $AUC_{0-\infty}$ (h·µg/mL) | $V_{ss}$ (mL/kg) | CL (mL/h/kg) | $MRT_{0-t}$ (h) | $C_{max}$ (µg/mL) | $T_{max}$ (h) |
|---|---|---|---|---|---|---|---|---|
| 0.1 | 25.50 ± 9.65 | 62.70 ± 25.60 | 71.10 ± 30.10 | 47.30 ± 6.38 | 1.62 ± 0.65 | 22.60 ± 7.54 | 2.65 ± 0.40 | 0.083 ± 0.00 |
| 1 | 50.50 ± 10.40 | 1420.00 ± 602.00 | 1510.00 ± 554.00 | 58.20 ± 22.00 | 0.74 ± 0.27 | 62.90 ± 11.30 | 27.80 ± 6.46 | 0.083 ± 0.00 |
| 10 | 63.40 ± 40.90 | 13,900.00 ± 2850.00 | 15,100.00 ± 2970.00 | 64.10 ± 23.70 | 0.69 ± 0.13 | 73.10 ± 8.05 | 240.00 ± 47.90 | 0.40 ± 0.78 |

$t_{1/2}$ effective half-time, $AUC_{0-t}$ area under the serum drug concentration-time curve up from time zero to the last measurable concentration, $AUC_{0-\infty}$ area under the serum concentration-time curve from time zero to infinity, which was calculated from the terminal phase data harvested; $V_{ss}$ apparent volume of distribution in the steady state, CL total body clearance following vascular administration of the drug, $MRT_{0-t}$ mean reaction time from time zero to the last measurable concentration, $C_{max}$ maximum drug concentration observed in the serum, $T_{max}$ time of the first occurrence of $C_{max}$.

utomilumab[31,32]. We demonstrated this effect by introducing Fc function into our CD137 mAb. However, Fc-FcR interactions may also lead to T cell depletion via NK cell-mediated effector function (ADCC) or macrophage-mediated phagocytosis (ADCP). Instead, we demonstrated that IBI319 requires its anti-PD-1 arm to achieve crosslinking to produce the agonistic effect on CD137 and that the presence of PD-1 is essential. Moreover, since IBI319 is a strong PD-1 binder (1137 times higher affinity for PD1 than for CD137), this design may localise most IBI319 molecules to microenvironments rich in PD-1-expressing T and NK cells, such as tumours and the TDLNs, and minimise systematic circulation, which may help limit off-target side effects.

Another interesting discovery is that the combined administration of αPD-1 and αCD137 mAbs, both with silenced Fc function, led to liver inflammation. Based on our data combined with previous reports, we propose the following potential mechanism for liver toxicity: upon PD-1 blockade, hyper-activated, nonspecific T cells in the circulation mediate the activation of liver macrophages, which then cause cytokine release as a secondary effect, leading to liver toxicity. If CD137 agonists harbour Fc function, this effect may be further enhanced by activating FcRs on liver macrophages[29]. Therefore, tumour-specific localisation is again essential in reducing the levels of circulating molecules, thus minimising toxicity. A few other molecules were reported to implement similar designs. PRS-343 (HER2/CD137, Pieris Pharmaceuticals) and two molecules from Roche (FAP/4-1BBL and CD19/4-1BBL) combine a tumour-associated antigen (TAA) with a CD137 agonist or 4-1BBL[8,33], but their single-agent efficacy might be limited since signal 1 (TCR signalling), the primary signal to activate T cells, is missing. Thus, for these molecules, combination therapy with PD-D/PD-L1 inhibitors or other agents, such as CD3 bispecific T cell engagers, could potentially activate signal 1. In addition, a few PD-L1/CD137 bispecific antibodies are currently under clinical development and are similar to IBI319[34,35]. PD-L1/CD137 bispecific antibodies may have advantages in tumour targeting due to the broad PD-L1 expression on tumour cells. We eventually chose PD-1 over PD/L1 for inclusion in IBI319, as targeting PD-1 blocks both the PD-L1 and PD-L2 and thus may lead to better T cell activation[36,37]. PD-L2 is a second ligand for PD-1 and is widely expressed by various tumours17. In addition to tumours, DCs and macrophages that localise in the lymphoid organs or even normal tissues express PD-L1. Therefore, a PD-L1/CD137 bispecific antibody may mediate extended immune synapse formation between T cells and DCs, leading to accelerated T cell exhaustion[38]. PD-L1 expression on normal tissues could also be a potential risk for unexpected side effects. In addition, we'd like to directly provide 4-1BB co-stimulation on PD-1 + T cells, with the aim to extend the durability of T cell reinvigoration and hopefully memory formation. Future efficacy and safety data generated in clinical studies may further reveal the differences in combining CD137 with PD-1 or PD-L1.

Indeed, based on the mode of action of IBI319, the clinical development of this agent should be carefully considered. As a single-agent, IBI319 could be primarily applied to patient populations with high TILs but low/no response to PD-1/PD-L1 therapies[39]. Furthermore, to take advantage of the T and NK cell-activating function of IBI319, another strategy is to explore combination treatments with T cell engagers (CD3 bispecific antibodies) that can enhance PD-1-positive TILs or antibodies with ADCC function to further enhance NK cell functions.

In summary, we have developed a CD137/PD-1 bispecific antibody, IBI319, with an improved therapeutic index that allows treatment of human cancer while avoiding potential toxicity (Fig. 5). We suggest further clinical development of this molecule

**Table 2 Main toxicokinetic (TK) parameters of IBI319 after multiple intravenous injections at 1, 10 or 50 mg/kg (mean ± SD, $n = 10$).**

| Dose (mg/kg) | Time | $AUC_{0-t}$ (h·μg/mL) | $C_{max}$ (μg/mL) | $T_{max}$ (h) |
|---|---|---|---|---|
| 1 | Day 1 | 1405.00 ± 259.33 | 26.83 ± 2.77 | 0.41 ± 0.84 |
| | Day 22 | – | – | – |
| 10 | Day 1 | 15,460.00 ± 23,663.70 | 245.80 ± 57.31 | 1.01 ± 2.53 |
| | Day 22 | 542.11 ± 358.75 | 104.20 ± 50.32 | 0.02 ± 0.00 |
| 50 | Day 1 | 80,710.00 ± 20907.81 | 1157.00 ± 174.11 | 0.02 ± 0.00 |
| | Day 22 | 21,743.00 ± 22429.54 | 1083.90 ± 584.71 | 0.02 ± 0.00 |

*AUC0-t* area under the serum drug concentration-time curve up from time zero to the last measurable concentration, *Cmax* maximum drug concentration observed in the serum, *Tmax* time of the first occurrence of Cmax. -: not calculated as the concentration was below detection limit.

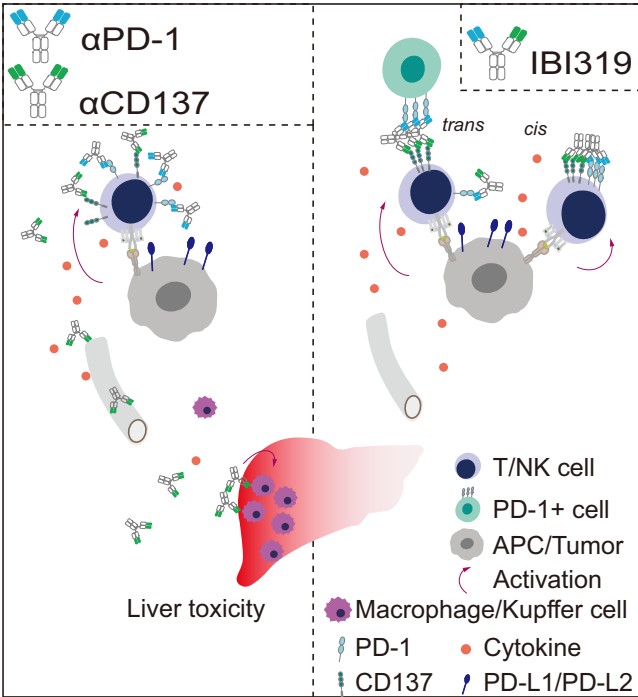

**Fig. 5 Working model of IBI319.** IBI319 activates CD137 in a manner restricted to PD-1-rich microenvironments, such as tumours and tumour-draining lymph nodes, limiting potential liver toxicity caused by the αCD137 and αPD-1 combination.

in PD-1-high and PD-1 blockade-nonresponsive or recurrent patients.

## Methods

**Antibodies, proteins, cells and mice**. Plasmids encoding the heavy and light chains (HC and LC) of IBI319, αCD137, αPD-1, αCD137/Hel, αCD137/Hel-Fc and αPD-1/Hel were transfected and expressed in Expi293F (ThermoFisher Scientific) or ExpiCHO-S cells (ThermoFisher Scientific). To ensure correct heterodimerization and HC-LC pairing of bispecific antibodies, various mutations, as previously reported, were introduced into the HC and LC and the IgG1_CH3 domain[40,41]. Briefly, the mutations were as follows for IBI319 (Kabat numbering): Q39Y, S131C, K218D, C220G, E356G, E357D, S364Q and Y407A in anti-CD137_HC; Q38R, A1D, K42D and D122K in anti-CD137_LC; Q39K, K62E, H168A, F170G, Y349S, T366M, K370Y, K409V in anti-PD-1_HC; and D1R, Q38D, L135Y and S176W in anti-PD-1_LC. In addition, L234A, L235A and N297Q (Kabat numbering[42]) were introduced in the Fc region to eliminate the effector functions of IBI319, αCD137 and αCD137/Hel. The antibodies used for flow cytometry are listed according to the antigen recognised (fluorophore, clone name, provider): hCD4 (BV570, RPA-T4, BD), hCD4 (BV510, UKT4, Biolegend), hCD8 (BV650, RPA-T8, BD), hCD137 (PE, 4B4-1, BD), hPD-1 (BUV737, EH12.1, BD), mCD45 (PE-Cy7, 30-F11, Biolegend), mCD4 (AF488, RM4-5, Biolegend), mCD3 (AF700, 17A2, Biolegend), mCD8a (BUV563, 53-6.7, BD), mNK1.1 (APC-Cy7, PK136, Invitrogen), mCD11b (BUV395, M1/70, BD), hIgG Fc (PE, M1310G05, Biolegend), hPD-1 (PE, 4B4-1, BD) (for QIFIKIT analysis). The antibodies used for IHC are listed according to the

antigen recognised (clone name, provider): mCD45 (D3F8Q, Cell Signaling Technology), mF4/80 (D2S9R, Cell Signaling Technology). A detailed antibody list including catalogue numbers and dilutions is provided in Supplementary Table 2.

The recombinant proteins used for binding affinity tests were hPD-1-his (PD1-H5221, Acro Biosystems), hCD137-his (41B-H5227, Acro Biosystems), hCD137L-Fc (41L-H5257, Sino Biological), biotinylated hFcRI (10256-H27H-B-20, Sino Biological), biotinylated hFcRIIa (CD-H82E7, Acro Biosystems), biotinylated hFcRIIb (CDB-H82E0, Acro Biosystems), biotinylated hFcγRIIIa (F158) (CDA-H82E8, Acro Biosystems), biotinylated hFcγRIIIa (V158) (CDA-H82E9, Acro Biosystems), biotinylated hFcγRIIIb (CDB-H82E1, Acro Biosystems), biotinylated hFcRn (FCM-H82W4, Acro Biosystems), hC1q (A099, Complement Technology), hPD-1-Fc (PD1-H5257, Acro Biosystems), hPD-L1-his (PD1-H5229, Acro Biosystems), mPD-1-Fc (PD1-M5259, Acro Biosystems), and mPD-L1-his (PD1-M5220, Acro Biosystems).

Jurkat-CD137-NFκB-Luc (J2332), Jurkat-PD-1-NFAT luciferase and CHOK1-PD-L1 (J1252) cells were purchased from Promega and cultured in the recommended medium according to the manufacturer's instructions. CHO-S-PD-1 cells expressing full-length human PD-1 (NM_005018.3) and Jurkat-CD137-NFκB-Luc2P-PD-1 cells expressing full-length human CD137 (NP_001552.2), an NFκB-Luc2P reporter (cloned from pGL4.32-luc2P/NF-κB-RE/Hygro, E8491, Promega) and full-length human PD-1 (NM_005018.3) were generated in-house by lentiviral transduction. CHO-S-PD-1 cells were cultured in CD FortiCHO medium (A11483-01, Gibco) supplemented with 1 μM methotrexate hydrate (M8407, Sigma) and 1× anticlumping agent (01-0057DG, Gibco). Jurkat-CD137-NFκB-Luc2P-PD-1 cells were cultured in RPMI 1640 medium (22400-089, Gibco) supplemented with 10% foetal bovine serum (FBS; SH30406.05, HyClone), 200 μg/ml hygromycin B (10687010, Invitrogen), 1 mM sodium pyruvate (11360-070, Gibco), 1% nonessential amino acids (11140-050, Gibco) and 500 μg/mL geneticin (10131-027, Gibco). Human PBMCs were purchased from Allcells and the informed consent was obtained from all donors. Human PBMCs were cultured in RPMI 1640 medium supplemented with 10% FBS.

The murine CT26 colorectal cancer cell line overexpressing human PD-L1 (CT26-hPD-L1) and BALB/c-hPD-1/hCD137/hPD-L1 mice (BALB/cJGpt-Pdcd1$^{em1Cin(hPDCD1)}$ Cd274$^{tm1Cin(hCD274)}$Tnfrsf9$^{em1Cin(hTNFRSF9)}$/Gpt, T007056) were obtained from GemPharmatech Co., Ltd. The murine MC38 cell line was obtained from OBiO Technology (Shanghai) Co., Ltd. C57BL/6-hPD-1/hCD137 mice (C57BL/6-Pdcd1$^{tm1(PDCD1)}$Tnfrsf9$^{tm1(TNFRSF9)}$/Bcgen, 120516) were purchased from Biocytogen Co., Ltd.

**Bio-layer interferometry (BLI) and surface plasmon resonance (SPR)**. Binding kinetics and affinities were measured by BLI and SPR. Briefly, for SPR, anti-Human Fc antibody (Ab97221, Abcam) was coupled on CM5 sensor chip (29-1496-03, GE Healthcare) to capture tested antibodies. Serially diluted antigens were then used to detect the binding affinities to tested antibodies using Biacore equipment (T200, GE Healthcare) and results were analyzed by Biacore T200 software (V 3.1, GE Healthcare). BLI was tested using Octet system (Octet RED96e, Sartorius). Recombinant proteins were diluted in SD buffer (Phosphate Buffered Saline (PBS) with 0.1% BSA and 0.05% Tween-20) to suitable concentrations. Antibodies or biotinylated recombinant proteins were loaded onto antihuman Fc-treated (AHC, Fortebio) or streptavidin (SA) biosensors, respectively. After loading, sensors were dipped into buffer containing antigen and then dissociated in SD buffer. For ligand competition experiment, human CD137 was loaded onto SA sensors then incubated with IBI319, αCD137, Urelumab, Utomilumab, and human CD137 ligand (CD137L), followed by CD137L. The data were analyzed with Fortebio data analysis 11.0 software and fit globally to a 1:1 binding model to determine the monovalent $K_D$.

**Antibodies binding to primary human T cells**. CD3 + T cells were isolated from human PBMCs using a pan T cell isolation kit (130-096-535, Miltenyi Biotec) according to the manufacturer's instructions. T cells were activated with human CD3/CD28 beads (Dynabeads®, 11131D, Gibco) (ratio of beads to cells = 1:1) at 37 °C for 4 days. After removing the beads with a magnetic shelf, collected

CD3 + T cells were transferred to a 96-well V-bottom plate (2 × 10⁵ cells/well). Each well was then re-suspended with 50 μL of FACS buffer (PBS with 2% FBS and 2 mM EDTA) containing an anti-hCD4 antibody, an anti-hCD8 antibody, a reagent from the LIVE/DEAD™ Fixable Near-IR Dead Cell Stain Kit (DCM; L34975, Invitrogen) and serially diluted tested antibodies, and the plate was incubated at 4 °C for 20 min. A secondary antibody, antihuman IgG Fc, was added for 20 min at 4 °C. The PD-1 and CD137 expression levels on T cells were determined in a separate tube. FACS acquisitions were performed on a FAC-Symphony™ A3 (BD Biosciences), and data were analysed with FlowJo software.

**Antibodies binding to the Jurkat-CD137 cell line.** Jurkat-CD137 cells were stained with Cell Trace™ Violet (CTV, C34557, Invitrogen), and CHO-S-PD-1 and CHO-S cells were stained with Cell Trace™ carboxyfluorescein succinimidyl ester (CFSE; C34554, Invitrogen). Stained Jurkat-CD137 cells were mixed with CHO-S-PD-1 or CHO-S cells and transferred to a 96-well V-bottom plate with 2 × 10⁵ and 4 × 10⁵ cells per well, respectively. Each well was then re-suspended with 50 μL of FACS buffer containing serially diluted tested antibodies and DCM at 4 °C for 20 min, followed by secondary antibody staining and FACS analysis as described above.

**PD-1/PD-L1 blocking activity in a Jurkat NFAT-Luc reporter assay.** The PD-1/PD-L1 Blockade Assay System (J1252, Promega) was used following guidance provided by the manufacturer. Briefly, CHOK1-PD-L1-TCRa cells were plated in a white 96-well plate at 4×10⁴ cells/well in a 100-μL volume and cultured at 37 °C for 16 h. Antibodies were prepared at a 2× working concentration and with 1:4 serial dilutions in assay buffer (RPMI 1640 medium+1% FBS). Jurkat-PD1-NFAT-Luc cells were centrifuged and re-suspended in assay buffer at 1.25×10⁶ cells/ml. Then, the 2× concentrated antibodies and Jurkat-PD1-NFAT-Luc cells were mixed at a 1:1 volume ratio and incubated at 37 °C for 6 h. To measure the luminescence signal, 80 μL of Bio-Glo™ Reagent was added to each well at room temperature, and the plate was incubated for 15 min and then read by a SpectraMAX i3 (Molecular Devices). Results were then calculated following the formula: fold induction = relative light units (RLU) treatment/RLU medium alone control. EC₅₀ values were fit with GraphPad Prism 8.0 software.

**CD137 agonist activity in a Jurkat NFκB-Luc reporter assay.** The 4-1BB Bioassay System (J2332, Promega) was used following the manufacturer's instructions. To evaluate the *trans*-cell activation mediated by IBI319, Jurkat-CD137 cells were co-cultured with or without CHO-S-PD-1 cells to test CD137 agonist activity. In the presence of CHO-S-PD-1 cells, Jurkat-CD137 cells were adjusted to 1 × 10⁶ cells/mL and mixed with CHO-S-PD-1 cells at 2.0×10⁶, 2.0×10⁵ or 2.0×10⁴ cells/ml to generate CHO-S-PD-1: Jurkat-CD137 cell ratios of 2:1, 0.2:1 and 0.02:1, respectively. In the absence of CHO-S-PD-1 cells, Jurkat-CD137 cells were adjusted to 5×10⁵ cells/mL. In each well, 25 μL of 3× concentrated antibody was added to 50 μL of co-cultured cells, and the mixture was incubated at 37 °C for 6 h. To measure the luminescence signal, 75 μL of Bio-Glo™ Reagent was added to each well, and as described above, the luminescence signal was determined, and the results were calculated. To investigate *cis*-cell-mediated activation within one cell, Jurkat-CD137 or Jurkat-CD137-NFκB-Luc2P-PD-1 cells co-cultured with 1/16 of the amount of CHO-S-PD-1 cells were separated into single cells by the 3D cell culture medium FCeM Advance Preparation Kit (383-10111, Nissan Chemical Industries). In each well, 50 μL of 2× concentrated antibody was added to 50 μL of above-mentioned cells, and the mixture was incubated at 37 °C for 6 h. To measure the luminescence signal, 100 μL of Bio-Glo™ Reagent was added to each well, and as described above, the luminescence signal was determined, and the results were calculated.

**MLR assay.** CD14 + monocytes were purified from human PBMCs with a pan monocyte isolation kit (130-096-537, Miltenyi Biotec). Monocytes were cultured in RPMI 1640 complete medium (10% FBS) in the presence of 20 ng/mL GM-CSF (215-GM/CF, R&D Systems) and 10 ng/mL IL-4 (204-IL, R&D Systems) for 5 days to generate immature DCs (iDCs). iDCs were then cultured for another day with 100 ng/ml LPS (24391-1MG, Sigma) to generate mature DCs (mDCs). T cells were isolated from the PBMCs of an allogenic donor using a pan T cell isolation kit. A total of 2×10⁴ mDCs, 2×10⁵ allogenic T cells and tested antibodies were then mixed in each well of a 96-well U-bottom plate and incubated for another 5 days. The supernatants were collected, and IFN-γ was analysed using a human IFN-γ enzyme-linked immunosorbent assay (ELISA) kit (DY285B, R&D Systems) according to the manufacturer's instructions.

**T cell co-stimulation assay.** Primary CD3⁺ T cells were isolated from human PBMCs using a pan T cell isolation kit and cultured in RPMI 1640 complete medium (10% FBS). One microlitre/mL ImmunoCult™ Human CD3/CD28 T Cell Activator (10971, STEMCELL) was used to sub-optimally activate T cells at 37 °C for 24 h. Then, the T cells were transferred into a 96-well U-bottom plate at 2×10⁵ cells/well and co-cultured with CHO-S or CHO-S-PD-1 cells at 1×10⁵ cells/well and tested antibodies. After another 3 days of stimulation, the supernatants were collected and tested to measure the IFN-γ level using a human IFN-γ ELISA kit.

**ADCC assay.** To test the ADCC activity mediated by antibodies, target cells (CHO-S-PD-1 cells, 3×10⁴ cells/well; Jurkat-CD137 cells, 2×10⁴ cells/well), ADCC effector cells (Jurkat-FcγRllla-NFAT-Luc cells, G7102, Promega) (1.5×10⁵ cells/well) and antibodies were mixed in assay buffer and incubated at 37 °C and 5% CO2 for 6 h. Bio-Glo™ Reagent was added to each well, and the luminescence signal was then measured as described above.

**T cell memory recall assay.** PBMCs from donors that were previously infected with CMV were treated with a CMV peptide pool (3619-1, Mabtech) and antibodies (50 nM) for 4 days. The supernatants were collected, and IFN-γ was tested with a human IFN-γ ELISA kit.

**QIFIKIT cell-surface protein quantification assay.** To quantify PD-1 and CD137 expression on the cell surface, QIFIKIT (K0078, Agilent) was used following the manufacturer's instructions. Briefly, standard samples were prepared using Vial 1 (setup beads) and Vial 2 (calibration beads). Cells to be analysed were stained with PE-conjugated antihuman PD-1 or PE-conjugated antihuman CD137 and then with a FITC-labelled secondary antibody provided in the kit. The standard samples and cells were then analysed by FACS. Following the manufacturer's guidance, a standard curve was then calculated using the standard samples to calibrate the antigen number in the cell samples.

**In vivo antitumour efficacy and PK study with syngeneic mouse models.** All mouse-related experiments complied with relevant ethical regulations for animal testing and research, and were approved by the Animal Use and Care Committee of Innovent Biologics. Mice were housed in the animal centre of Innovent Biologics with specific pathogen-free (SPF) housing condition (light/dark cycle: 12 h/12 h; temperature: 22-23 °C; humidity: 55%). All animals were housed in the same room and randomly distributed into experimental/control groups (2–5 animals per cage). CT26-hPD-L1 (1×10⁶ cells/mouse) or MC38 (1×10⁶ cells/mouse) tumour cells in 200 μL of PBS solution were subcutaneously (s.c.) injected into the right flank of BALB/c-hPD-1/hCD137/hPD-L1 mice (source: GemPharmatech-T007056, female, 6-8 weeks old, stock number: 20200210001 for experiments in Fig. 3a and Supplementary Fig. 3c) or C57BL/6-hPD-1/hCD137 mice (source Biocytogen-120516, female, 5–6 weeks old, stock numbers: 2019102101N2 and 2019102801N2 for experiment in Fig. 3b and supplementary Fig. 3d–f; 2021030101N3 for experiment in Fig. 3e and 2021030101N2 for experiment in supplementary Fig. 3g), respectively. The longest axis (L) and the shortest axis (W) of the tumour were measured with callipers, and the tumour volume (V) was then calculated following the formula: V (mm³) = L×W²×0.5. Mice were randomised when the mean tumour volumes reached an average of 130 mm³ (6-7 animals per group). Antibodies were diluted in PBS and intraperitoneally (i.p.) administered once per week 3 times. Body weights and tumour volumes were measured twice per week. The TGI rate was calculated using the formula: TGI (%) = 100%× (V_control − V_treated)/(V_control −V_pretreated) (V_control: tumour volume of the control group, V_treated: tumour volume of the treated group, V_pretreated: tumour volume before treatment). TILs were isolated using anti-CD45 beads, and immune cells were analysed using a FACSymphony™ A3 (BD Biosciences).

For the PK study, a single-dose PK study was performed with C57BL/6-hPD-1/hCD137 mice intravenously injected with IBI319 at 10 mg/kg. Blood samples were collected and processed for serum isolation at 8 time points (5 min, 30 min, 2 h, 6 h, 24 h, 48 h, 96 h and 168 h postdose). The serum levels of IBI319 were measured using an in-house-developed ELISA with human PD-1 (ACRO, PD1-H5221) as the capture reagent and biotinylated human CD137 (ACRO, 41B-H82F7) as the detection reagent. Serum concentration-time profiles were used to estimate PK parameters using non-compartmental analysis (PK Solver 2.0).

**FACS gating strategy.** All FACS sequential gating strategies for the analyses of targeted cell populations are indicated in Supplementary Fig. 7.

**IHC.** After animal euthanasia (carbon dioxide exposure followed by cervical dislocation), the liver, tumour and TDLNs were collected immediately, fixed in 10% neutral buffered formalin and processed for paraffin embedding. Paraffin sections were prepared using a microtome and stained using haematoxylin (51275, Sigma) and eosin (HT110316, Sigma) (H&E) or IHC. Images were scanned with an Aperio Versa 8 (Leica Biosystems). CD45 + and F4/80+ cells were quantified with Halo Image Analysis software (Indica Labs).

**scRNA-seq of TILs.** Tumour samples from MC38 tumour-bearing C57BL/6-hPD-1/hCD137 mice administered antibodies were collected at the end of the experiment. Tissues were dissociated into single-cell suspensions by a tumour dissociation kit (130-096-730, Miltenyi Biotec). scRNA-seq libraries were constructed using the Chromium Single Cell 3' Reagent Kit (V3) according to the protocol provided by the manufacturer. The CellRanger (10X Genomics, 2.1.1 version) analysis pipeline was used for sample demultiplexing, barcode processing, and gene counting based on a mouse genome reference sequence (GRCm38). Seurat (v3.1.2) (https://satijalab.org/seurat/) was used for quality control and downstream analysis. Cells with fewer than 500 genes or over 10% of the genes derived from the mitochondrial genome were removed. Potential residual red blood cells were also

filtered out based on the expression of haemoglobin genes. Doublets predicted by Scrublet[43] were also removed. Only tumour-infiltrating immune cells with CD45 expression were selected for further analysis. The single-cell data of different samples were first separately normalised using the SCTransfrom method and then integrated by anchor genes to further perform dimensionality reduction (Seurat function RunUMAP), clustering (Seurat function FindClusters) and visualisation.

**PK and toxicity study of IBI319 in cynomolgus monkeys**. All cynomolgus monkey-related experiments were conducted at WestChina-Frontier PharmaTech Co., Ltd. (WCFP) in accordance with standard operating procedure and were complied with relevant ethical regulations. The experiments were approved by the Animal Care and Use Committee of WCFP. The 4-week repeated-dose toxicology study was performed in compliance with the principles of national medical products administration (NMPA), food and drug administration (FDA) and organisation for economic cooperation and development (OECD) good laboratory practice (GLP). The animal source was Hainan Jingang Biotech Co., Ltd. with the batch number of 460012000000369 for both the single-dose PK study and the 4-week repeated-dose toxicity studies. Briefly, in the single-dose PK study, cynomolgus monkeys were intravenously injected with IBI319 at doses of 0.1, 1 and 10 mg/kg (three males and three females in each group). Serum samples for drug concentration and RO analyses were collected from all animals before dose administration and at 5 min; 2, 4 and 8 h; and 1, 2, 3, 7, 10, 14, 21, 28 and 35 days postdose. The serum levels of IBI319 were determined using the protocol described for the mouse experiments. The RO on T cells was confirmed using competitive ELISA by adding biotinylated αPD-1 (parental antibody of IBI319). In the 4-week repeated-dose toxicity study, IBI319 was administered weekly to cynomolgus monkeys via i.v. infusion at 1 mg/kg, 10 mg/kg, or 50 mg/kg for a total of 5 doses (5 males and 5 females in each group), followed by a 4-week recovery phase. After the 5th dose, three monkeys/sex/group (main group) were euthanized and necropsied, and the remaining two monkeys/sex/group were observed for an additional 28 days prior to being euthanized. The following parameters were examined during the study: viability, clinical observations (including at the injection site), food consumption, body weight, body temperature, safety pharmacology (electrocardiography, blood pressure, heart rate and respiratory rate), ophthalmology, clinical pathology (haematology, clinical chemistry and urinalysis), immunology [complement C3/C4, circulating immune complex, immunotyping (CD3/4/8 + T cells, B cells, and NK cells), immunoglobulins (IgG, IgA and IgM)], immunogenicity (ADA analysis), toxicokinetics, bone marrow smear, organ weights and ratios, gross pathology and histopathology. The following PK parameter abbreviations are used in Table 1 and Table 2: $t_{1/2}$, effective half-time; $AUC_{0-t}$, area under the serum drug concentration-time curve up from time zero to the last measurable concentration; $AUC_{0-\infty}$, area under the serum concentration-time curve from time zero to infinity, which was calculated from the terminal phase data harvested; $V_{ss}$, apparent volume of distribution in the steady state; CL, total body clearance following vascular administration of the drug; $MRT_{0-t}$, mean reaction time from time zero to the last measurable concentration; $C_{max}$, maximum drug concentration observed in the serum; and $T_{max}$, time of the first occurrence of $C_{max}$.

**Statistical analysis and reproducibility**. Statistical methods are described in the corresponding figure legends and p values are shown in the figures. All t-tests are two sided. All experiments except the in vivo studies were repeated at least twice or using at least three PBMC donors.

**Reporting summary**. Further information on research design is available in the Nature Research Reporting Summary linked to this article.

## Data availability

The source data of plots in figures are provided in Source Data file. The single-cell RNA sequencing data generated in this study have been deposited in figshare.com under the hyperlink of https://figshare.com/s/af514391833ca9622e27 and DOI of https://doi.org/10.6084/m9.figshare.16610875.v1. The dataset named "data.submit.rds" could be downloaded directly without access code. All other data generated during the current study are available from the corresponding authors upon reasonable requests. Source data are provided with this paper.

## Code availability

No custom code was generated for this study.

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

## Acknowledgements

The study was fully sponsored by Innovent Biologics and the molecule IBI319 was initiated from a collaboration with Eli Lilly co. We thank Genergy Biotechnology (Shanghai) Co., Ltd. for scRNAseq library construction and sequencing.

## Author contributions

Y.Q., Y.Q., J.D., X.W. and W.S. carried out the in vitro experiments and analysis. N.L., M.Z. and D.W. carried out the no-human primate PK and toxicity evaluation and analysis. J.S. conduced bioinformatics. H.W. and Y.W. conducted IHC experiments and analysis. T.S., Y.L. and BC conducted mouse experiments. X.L., F.G. and Z.W. conducted protein binding assays. X.W. and W.X. conceived of the presented idea and supervised the project.

## Competing interests

All authors were employees of Innovent Biologics (Suzhou) Co. when the works of this report were conducted.
