## [Peer Review File · Nature Communications]

Cancer immune therapy with PD-1-dependent CD137 co-stimulation provides localized tumour killing without systemic toxicityREVIEWER COMMENTS

Reviewer #1 (Remarks to the Author):

The manuscript by Qiao et al describes a new CD137/PD-1 bispecific antibody (IBI319) able to function as an agonist for CD137 and simultaneously block PD-1/PD-L1 interactions. In a series of well-conducted experiments they showed that IBI319 can provide potent antitumor activity in two different colon cancer models developed in mice having humanized CD137 and PD-1 receptors. Interestingly, this antitumor effect was stronger than the one obtained with monospecific antibodies and similar to the combination of antiCD137 and anti-PD-1 antibodies, but in contrast to this last approach it did not induce liver toxicity. Lack of toxicity was also assessed in cynomolgus monkeys, which further supports the clinical development of this new bispecific antibody, especially since anti-CD137 mAbs, like urelumab, have shown high hepatic toxicity in patients. The authors also showed that the IBI319-mediated activation of CD137 required the presence of PD-1 either in different cells or in the same cells. The mechanism by which binding to PD-1 favors CD137 activation is not completely clear but the authors hypothesized that it could help promote trimerization of CD137 on target cells, although this was not assessed. Overall, this is a relevant study with clinical potential. However, some points must be clarified before publication:

Major points

- Fig 2E. Why do the authors normalize IFN γ levels to IgG1? Please explain. Also please show PD-1 levels in T cells from the experiment represented in Fig 2D and 2E.
- Fig 3B. In the MC38 model tumor and immune cells are expressing murine PD-L1 in contrast to the CT26—hPD-L1 model. That means that most likely inhibition due to PD-L1 expressed in the tumor is not taking place in this model. Maybe the authors could add a short comment about this.
- Fig S5. Why do you show ALT/AST ratios? Please provide separate data for AST and ALT.
- Fig 4B It is not clear for this reviewer to which time points in the experiment day 1 and 22 correspond... 1 and 22 days after the fifth dose? Please clarify. Since authors are giving five doses of IBI219 at weekly intervals it would be interesting to show their levels at different time points during the experiment and not only after the five doses.
- Finally, regarding the mechanism, can the authors provide any indication that IBI219 is mediating CD137 trimerization on target cells?

Minor points

- Cite figures in order...For example: Fig S3 is mentioned before Fig S2; Fig. 2B is introduced in the text before Fig.2A; Fig 5 is cited before Fig 4... In addition Fig. S3A is not cited in the main text.
- Line 1: This sentence needs to be rewritten, it is not easy to understand: "Nevertheless, to simultaneously reduce the off-target toxicity while remaining anti-tumor efficacy are the remaining challenges to advance CD137 agonists in clinical applications, and to solve which the Fc function, affinity and binding epitope shall be all considered in a reasonable manner for the desired new molecule.
- Fig S1A. The colors of the lines in the graphs are not easy to distinguish...maybe the figure could be improved by using thicker lines.
- Line 145: The authors say that "A slightly weaker PD-1/PD-L1 blocking activity was observed for IBI319 (EC₅₀ 8.08 \pm 1.51 nM) compared to α PD-1 mAb (EC₅₀ 1.09 \pm 0.24 nM)" I think a difference of almost one log (8-fold) in blocking activity is not a slight difference... Please change this sentence stating that IBI39 has a weaker inhibitory effect than α PD-1 mAb.
- Line 170. Please define here MLR

- Line 178 "Similarly, using primary T cells that were activated by small amount of CD3/CD28 T cell activator (1 μ L/mL)". Please indicate proper units (1 μ L does not indicate how much of the activator was used).

- Line 195. The authors say that they chose: "a moderate responsive model CT26 and a good responsive model MC38." referring to the PD-1 response. However, according to the reference they provide (ref. 22) it is the opposite: CT26 responds well to anti-PD-1 therapy but MC38 does not respond...

- Fig. S3C. In line 210 it is stated that "In consistent with IBI319's efficacy, both the PD-1 and CD137 were up-regulated on MC-38 tumor infiltrated T cells measured by FACS (Fig. S3C)." However I do not see a clear upregulation of CD137 in TILs compared to PBLs...Please modulate this statement or show data supporting this. Also in this figure the last two lines correspond to figure S4 legend...(they are duplic

Reviewer #2 (Remarks to the Author):

In this manuscript entitled "PD-1 dependent CD137 co-stimulation as a novel strategy to enhance efficacy while avoiding off-tumor toxicity", Qiao et al. designed and generated a CD137/PD-1 bispecific antibody (IBI319) which has higher affinity to bind to PD-1 and lower affinity to bind to CD137 to induce restrictive activation of CD137 in PD-1 rich microenvironment. Later, the authors conducted in-vitro cell based experiment and showed that the CD137 activation is dependent on the presence of PD-1. In addition, in vivo mouse model experiments were carried out to characterize and investigate IBI319's activity and pharmacokinetics. The mouse model data showed that IBI319 has very promising antitumor activity with limited hepatotoxicity indicated by CD45 and F4/80 markers. Furthermore, preclinical study of pharmacokinetic and safety profile of IBI319 were also conducted in cynomolgus monkeys. Overall the study is well designed and conducted. However, some issues need to be addressed:

1. This is an important area. Would put the recent publication <https://pubmed.ncbi.nlm.nih.gov/33310772/> emphasizing irAE are important to avoid. Add it is important to get efficacy without irAE and how this molecule plays a role. Also refer to the recent publication <https://jitc.bmj.com/content/8/2/e001650> why 41BB is important for this pathway.
2. IBI319 was designed to possess relatively low affinity to bind to CD137 and therefore favor strong binding to PD-1. Fig 1C showed that IBI319 could compete and completely block the binding of CD137 ligand (CD137L) to CD137. In this Octet RED96e system binding kinetics and affinity assay, what's the binding affinity of natural CD137L to CD137? Does IBI319 exhibit higher affinity than natural CD137L?
3. Figure 3D showed the ratio of different cell clusters in each treatment groups, was there any data showing the absolute cell clusters numbers/amounts to compare each cell clusters between treatment groups?
4. Currently a few PD-L1/CD137 bispecific antibodies are under clinical development. What's the rational to choose PD-1 over PD-L1/L2 to design IBI319 in this study? Have the authors compared IBI319 with other PD-L1/CD137 bispecific antibodies, whether the other PD-L1/CD137 bispecific antibodies' CD137 activity depend on PD-L1/L2?
5. What's the current status of IBI319? The preclinical data seems very promising, is it under clinical development? It would nice to see path forward.

Reviewer #3 (Remarks to the Author):

This is an interesting publication introducing a novel 1+1 PD1/CD137 bispecific antibody with an interesting design: High PD-1 affinity and low CD137 affinity. This is to my knowledge the first

report on a PD-1 targeted CD137 agonist and the 1st BsAb targeting PD-1 and CD137 in clinical trials working through PD-1 inhibition and subsequent cis/trans 4-1BB activation. The studies are generally well performed and insightful, however, some experiments lack suitable controls.

From the experiments one remaining question is whether the dose response of PD-1 inhibition on the one hand and CD37 agonism on the other hand match to each, or in other words, is the dose required for PD-1 blockade also the optimal dose to agonize CD137. This could only be shown comparing to a combination of maximal PD-1 inhibition and maximal CD137 agonism, however, for that purpose e.g. the use of Fc dependent CD137 antibodies is required.

The authors should 1) try to use more physiologically relevant T cell models in vitro/ex vivo and 2) in the in vivo studies rather compare to an Fc active CD137 antibody that mediates stronger liver tox but also stronger efficacy. The concern with the in vivo studies as they are is that either PD-1 + CD137 is not tested (CT26) (and thus it is not shown how good agonizing both pathways can be) or the superiority of combining PD-1 and CD137 over the single agents is not shown (MC38).

Similarly, in the MLR assay the BsAb is not superior to PD-1 monotherapy, but just to the combination, which weakens the findings reported.

Generally, if these point can be addressed having a bispecific antibody that induces CD137 agonism through PD-1 binding and independent of FcγR is of course advantageous and a very good approach. Particularly, as conventional 1st gen CD137 agonists have been reported to be associated with liver tox I am feeling that the current data set still lacks some elements to unequivocally demonstrate the CD137 contribution in the BsAb to checkpoint inhibition. In general, but maybe not as a requirement for this paper, it would also be nice to show superiority or at least equivalency of the BsAb over the combination of a PD-1 antibody with 2nd gen 4-1BB agonists that lack liver tox or e.g. PD-L1 targeted CD137 agonists. Ultimately, given the tox of conventional CD137 agonists, showing that the BsAb is equally efficacious as the combination would be sufficient, e.g. for the BsAb not necessarily better efficacy needs to be shown. Another important question is whether the affinity chosen for the CD137 arm is optimal, or whether better CD137 agonism could be achieved using a slightly higher affinity antibody e.g. 50 or 100 nM would be advantageous given the relatively low levels of PD-1 available for crosslinking

Last but not least, the text should be more thoroughly checked by a native English speaker, I included some examples below

General points

Include details on design and production of the BsAb, how is correct light chain association achieved? In vitro assembly of the BsAb?

What is the expression level of PD-1 on CHO-S-PD-1 and how does it compare to natural PD-1 expression, this may be important to show the relevance as in general PD-1 expression is low and the question is how much 4-1BB agonism can be achieved via crosslinking through PD-1

Figure 1E following: Considering, that you see hardly binding to 4-1BB with the aCD137 (red) antibody it is reasonable to assume that you have hardly any 4-1BB expressed on this cells (of course to claim this a high affine aCD137 control is missing here). Furthermore, seeing than the strong MFI increase (with a increase in EC50) of the bispecific IBI319 (green) compared to aPD-1 (blue) one could also assume that the difference is due to the bivalent to monovalent binding to PD-1 and not additive effect of bispecific binding. E.g. a high affine monovalent binder will double the MFI compared to a high affine bivalent Ab (as the double amount of Fc (Ab) will be clustered on the cell surface

Figure 2A, where does the weak activity of PD-1 antibodies come from? They should not affect NFκB signaling per se, is this just background?

This cell line monitor only NFκB and theirfore 4-1BB activity. But not the influence of PD-1

blockade. This could be made clearer (like Fold induction of NFkB activation).

Does Jurkat-CD137-NFkB-Luc2P-PD1 display the same CD137 expression as Jurkat-CD137-NFkB-Luc2P?

Does Jurkat-CD137-NFkB-Luc2P-PD1 display the same PD1 expression as CHO-S-PD1?

Clarify:

1) Jurkat-CD137-NFkB-Luc2P and the Jurkat-CD137-NFkB-Luc2P-PD1 display the same activation potential (like stimulation with TNFa)

2) Jurkat-CD137-NFkB-Luc2P and the Jurkat-CD137-NFkB-Luc2P-PD1 display the same expression of CD137

3) Jurkat-CD137-NFkB-Luc2P-PD1 and CHO-S-PD1 display the same expression of PD1?

Fig 2C: The picture is a little bit misleading as it implies that you did not have seeded the same amount of Jurkat-CD137-NFkB-Luc2P (4 light dots) and the Jurkat-CD137-NFkB-Luc2P-PD1 (7 red dots). However for a fair comparison the cell number of the reporter cell lines should be the same. Please clarify

Fig 2D: The CD137 antibody used here is the Fc-dead version, if yes, it is a superagonist that works without FcγR binding? It seems that the MLR effect is driven by PD-1 inhibition, does the combination with CD137 add anything significant? This may be a consequence of CD137 being Fc dead? Actually, in all cases the BsAb is not significantly different from PD-1 alone -> conclusions that it is superior to the combination need to be adapted, there is sometimes significance compared to the combination, but not to single agent PD-1? So, in Fig 2D the BsAb appears not better than PD-1 alone, it is only if CHO-PD-1 is added, but not in a classical MLR. It would be better to show this via PD-1 expression on T cells superior MLR activity can be seen

In vivo studies: The CD137 antibody used here has no Fc function, correct? If an Fc dead CD137 antibody is used where does the liver tox come from, also an Fc dead one should not be efficacious? Does the antibody exhibit super-agonism? In general, from the data it appears the CD137 antibody is largely inactive regarding 4-1BB agonism? If this is true it may not be the ideal comparator and personally a Fc active CD137 antibody may be the better comparator to show that equal or better efficacy can be achieved without liver tox with the BsAb. Please clarify in the text the lack of Fc function

Fig 3A. The combination as a comparator is lacking; any explanation why it prevents relapse, outgrowth at later times, but early on is not superior to PD-1

Fig 3B: the comparison to PD-1 monotherapy alone in the MC38 model is missing, it could be that the efficacy comes just from PD-1 inhibition alone, and not from PD-1 + CD137 combination to which it is compared; is the BsAb really better than PD-1 in MC38? In the supplement PD-1 monotherapy data is shown but only for 3 mg/kg, not for the combination dose

Fig 3E: clarify in the text that CD137 mAb had no Fc function

Fig 4A. Explanation for drop of antibody levels around 240-360h; is this ADA related, explain

Discussion: A few other molecules were reported to implement similar designs, PRS-343 (HER2/CD137, Pieris Pharmaceuticals) and two molecules from Roche (FAP/4-1BBL and CD19/4-1BBL) combine a tumor associated antigen (TAA) with CD137 agonist or 4-1BBL, but their efficacy as a single agent might be limited since the TCR signal 1 to primarily activate T cell is missing. => discuss possibility to combine these agents with PD-1/PD11 inhibition to re-activate the signal 1

Wording/English points

in general: replace "remain" by "retain" throughout the text

The following sentences would need re-writing as there are errors or they are difficult to understand

Nevertheless, to simultaneously reduce the off-target toxicity while RETAINing anti-tumor efficacy are the remaining challenges to advance CD137 agonists in clinical applications, and to solve which the Fc function, affinity and binding epitope shall be all considered in a reasonable manner for the desired new molecule. (2nd sentence cannot be understood)

to its ligands PD-L1/PD-L2 provides a negative feedback signal against TCR activation via protein tyrosine phosphatase SHP-213, the inhibition of which via anti-PD-1 /PD-L1 antibodies is mainly through signal (something is missing in the 2nd sentence, also please note the role of CD20 co-stimulation in order for CPI to work)

Therefore, simultaneously blocking PD1/PD-L1 and activatING CD137

arm is from Sintilimab (α PD-1), an approved PD-1 blockade??

indicating an overlappING binding epitope

strong blocking OF PD-1 and appropriate agonizing OF CD137

and the induced PD-1/CD137 on PBMC is relatively low.

Therefore, in addition to confirm IBI319 in remaining this synergy, we ...

In consistent with IBI319's efficacy

Further, the bispecific format of IBI319 has shown it comparable efficacious as the combination of the two agents in in vivo setting.

Comments from Reviewers

Reviewer #1 (Remarks to the Author):

Major points

1.1 - Fig 2E. Why do the authors normalize IFN γ levels to IgG1? Please explain.

Response:

Fig. 2E shows the Relative IFN- γ production level in T cell co-stimulation assay. The reason of showing normalized data to IgG1 group instead of absolute IFN- γ concentration is that the basal IFN- γ production level varies among three PBMC donors. However, the activation trends upon antibody addition among donors were close. Therefore, in order to plot all three donors into one graph and show the difference among treatments, we presented the data as normalized to their baselines. Since PBMCs are considered as primary cells, we assume that individual donors can produce very different basal cytokines. The absolute IFN- γ values, with each PBMC donor plotted separately, were shown as Supplementary Fig. 2d and explained in text line 182. A similar normalization was applied to the MLR assay in Fig. 2d and the absolute values were incorporated into Supplementary Fig. 2c explained in text lines 172-173.

1.2 - Also please show PD-1 levels in T cells from the experiment represented in Fig 2D and 2E.

Response:

The PD-1 and CD137 levels in T cells of 3 other donors treated in the same condition as in Fig.2d or 2e were incorporated into Supplementary Fig. 2c-bottom and Supplementary Fig. 2d-bottom, respectively. Relevant text in Main body: lines168-170 (“T cells in the allogeneic incubation system upregulated PD-1 and CD137, enabling the analysis of functions related to PD-1 blockade and CD137 agonism (Supplementary Fig. 2c- bottom).”) and 177-180 (“Similarly, when primary T cells were activated by a small amount of CD3/CD28 T cell activator (1 μ L/mL), in which PD-1 expression was sub-optimally upregulated (Supplementary Fig. S2d- bottom”).

1.3 - Fig 3B. In the MC38 model tumor and immune cells are expressing murine PD-L1 in contrast to the CT26—hPD-L1 model. That means that most likely inhibition due to PD-L1 expressed in the tumor is not taking place in this model. Maybe the authors could add a short comment about this.

Response:

It's previously reported that murine PD-L1 (mPD-L1) binds to human PD-1 (hPD-1) with a similar affinity ($K_D \sim 8 \times 10^{-6}$ M by SPR, Lin et al., PNAS 2008, doi: 10.1073/pnas.0712278105.) as hPD-L1 binding to hPD-1 (K_D : 2.6×10^{-6} M by SPR, Li et al., Cancer Sci. 2018 doi: 10.1111/cas.13666). And via an in-house measurement by BLI, we confirmed that mPD-L1 binds to hPD-1 with K_D at 5.39×10^{-7} M. This affinity is comparable to mPD-L1 binding to mPD-1 (K_D : 1.6×10^{-7} M), suggesting that the hPD-1 expressed by the transgenic mice can bind to mPD-L1 expressed by MC38 cells. In addition, the C57BL/6-hPD-1/hCD137 transgenic model was generated in the way that the extracellular domains of PD-1 and CD137 were replaced by human proteins, and the intracellular domains responsible for signal transductions remained unchanged and functional. In a previous

study, PD-1 blockade inhibited the growth of MC38 model in C57BL/6-hPD-1 mice, which is the parental strain of C57BL/6-hPD-1/hCD137 mice (Zhang et al., *Antibody Therapeutics* 2018, <https://doi.org/10.1093/abt/tby005>). Therefore, taken together, MC38 inoculated in C57BL/6-hPD-1/hCD137 transgenic mice is an appropriate model to evaluate PD-1 inhibition function. The mPD-L1 and hPD-L1 sequence comparison and affinity results mentioned above were incorporated into Supplementary Fig. 3a and 3b, respectively. Relevant text in Main body: lines 223-226 (“Of note, it was previously reported that murine PD-L1 (mPD-L1) binds to human PD-1 (hPD-1) with an affinity similar to that of hPD-L1 binding to hPD-L1 (Supplementary Fig. 3a, b), suggesting that the hPD-1 expressed by transgenic mice can bind to mPD-L1 expressed by MC38 cells.”).

..

1.4- Fig S5. Why do you show ALT/AST ratios? Please provide separate data for AST and ALT.

Response:

We showed the ALT/AST ratios as most causes of liver cell injury are associated with a greater increase in ALT than AST thus the ratio of the two is usually used as a clinical indicator (Salaspuro, M. *Enzyme* (1987) 37, 87–107). The separated AST and ALT were incorporated into Supplementary Fig. 5c. Relevant text in Main body: lines 276-279 (“Furthermore, there were no significant increases in the alanine aminotransferase (ALT) or aspartate transaminase (AST) levels or the ratio of ALT/AST32 in the blood with IBI319 treatment or the α PD-1 and α CD137 mAb combination therapy (Supplementary Fig. 5c”).

1.5- Fig 4B It is not clear for this reviewer to which time points in the experiment day 1 and 22 correspond... 1 and 22 days after the fifth dose? Please clarify.

Response:

Thanks for pointing the obscure expression out. The time points of Day 1 and Day 22 associated to the multi-dosing experiment (Fig. 4c, d). Drug administrations were on Day1, Day8, Day15, Day22 and Day29, and the table in Fig. 4d (previous Fig. 4B) showed the main toxicokinetic (TK) parameters after the first and fourth dosing. Therefore, accordingly, Day1 and Day22 indicated the 1st (day1) and 4th dosing (day22), respectively. AUC_{0-t} meant the exposure within 7 days after the corresponding dosing time. To make the expression clear and precise, in the main body (line 305), “The multiple-dose PK/toxicokinetic (TK) study” was revised to “a multiple-dose toxicokinetic (TK) study”, and accordingly in Fig. 4 legend, “(B) Main PK/TK parameters of IBI319” was revised to “d Main TK parameters of IBI319”. Relevant text in Main body: lines 308-310.

1.6- Since authors are giving five doses of IBI219 at weekly intervals it would be interesting to show their levels at different time points during the experiment and not only after the five doses.

Response:

Thanks for the suggestion. We designed two 7-day blood sampling periods after Day 1 (1st dose) and Day22 (4th dose), respectively, which was sufficient for the TK study. It was not possible to do more frequent blood sampling due to animal ethics of GLP-toxicity study (some animals were sacrificed for histopathology 1 day after the 5th dose, so the TK sampling time was set after the 4th dose). The toxicokinetic (TK) profile after the 1st dose could tell the linearity and sex difference of the test article, while the profile after the 4th dose could tell the drug immunogenicity (blood concentrations and anti-drug antibody (ADA) titer) and drug accumulation after repeated dose, which could support the toxicity explanation. The blood concentrations after Day 1 (1st dose) and

Day 22 (4th dose) at each time point were incorporated into Fig 4c. Relevant text in Main body: lines 308-310.

1.7- Finally, regarding the mechanism, can the authors provide any indication that IBI219 is mediating CD137 trimerization on target cells?

Response:

Regarding the trimerization of CD137 upon antibody binding, we showed data previously (original data: Fig. 2b) and currently include a new result (Supplementary Fig. 2g) using a Jurkat-CD137-luc reporter assay: CD137/PD-1 bispecific antibodies induce an NFκB-Luciferase signal in a PD-1-dependent manner (Fig. 2a, Supplementary Fig. 2g). We assume that CD137 antibody mediated CD137 trimerization as it is well documented that CD137 belongs to the TNF super family which the ligand as well as the receptor trimerization are needed to trigger downstream signal (see references 4-6 in the manuscript): the recruitment of TNF receptor-associated factors (TRAF1, TRAF2 and TRAF3) and signaling cascades such as NFκB, ERK, JNK, p38 MAPK5. However, to specifically address IBI319 inducing trimerization, structure-based assays such as protein co-crystallization could be introduced in future studies. Relevant text in Main body: lines 34-38 and 134-165.

Minor points

1.8- Cite figures in order...For example: Fig S3 is mentioned before Fig S2; Fig. 2B is introduced in the text before Fig.2A; Fig 5 is cited before Fig 4... In addition, Fig. S3A is not cited in the main text

Response:

Thanks for pointing out. We have revised the Figure no. in the current manuscript in order and made sure that all figures are cited in the main text.

1.9- Line 80: This sentence needs to be rewritten, it is not easy to understand: "Nevertheless, to simultaneously reduce the off-target toxicity while remaining anti-tumour efficacy are the remaining challenges to advance CD137 agonists in clinical applications, and to solve which the Fc function, affinity and binding epitope shall be all considered in a reasonable manner for the desired new molecule.

Response:

Thanks for pointing out the obscure expression. Please find the current text in lines 56-59: "Nevertheless, reducing off-target toxicity while retaining antitumour efficacy is a continuing challenge in advancing CD137 agonists into clinical applications, and overcoming this issue will likely require consideration of the Fc function, affinity and binding epitope properties of the desired new molecule"

1. 10- Fig S1A. The colors of the lines in the graphs are not easy to distinguish...maybe the figure could be improved by using thicker lines.

Response:

We have improved the figure S1A by using thicker line: currently changed to Supplementary Fig. 1b).

1.11- Line 145: The authors say that "A slightly weaker PD-1/PD-L1 blocking activity was observed

for IBI319 (EC50 8.08 ± 1.51 nM) compared to αPD-1 mAb (EC50 1.09 ± 0.24 nM)” I think a difference of almost one log (8-fold) in blocking activity is not a slight difference... Please change this sentence stating that IBI39 has a weaker inhibitory effect than αPD-1 mAb.

Response:

We have revised the corresponding sentence in the current manuscript lines 132-133 “Weaker PD-1/PD-L1 blocking activity was observed for IBI319 (EC50 8.08 ± 1.51 nM) than for the αPD-1 mAb (EC50 1.09 ± 0.24 nM)...”

1. 12- Line 170. Please define here MLR

Response:

MLR is defined in the current manuscript line 166 ...mixed leukocyte reaction (MLR)...”

1. 13- Line 178 “Similarly, using primary T cells that were activated by small amount of CD3/CD28 T cell activator (1 μL/mL)”. Please indicate proper units (1 μL does not indicate how much of the activator was used).

Response:

The proper concentration was not disclosed by the supplier of the *of CD3/CD28 T cell activator*. According to the user manual 25 μL /mL is recommended to be the strong stimulation of T cells. We chose 1 μL/mL to stimulate T cells under a sub-optimal condition (relative lower PD-1/CD137 expression as compared to full stimulation (supplementary Fig. 2d-bottom, versus 1d). Regarding the CD3/CD28 antibody concentrations, according to patent “WO2016033690A1” which reported a similar product, the concentrations were 0.5 μg/ml and 0.5 μg/ml for CD3 and CD28 antibodies.

1.14- Line 195. The authors say that they chose: “a moderate responsive model CT26 and a good responsive model MC38.” referring to the PD-1 response. However, according to the reference they provide (ref. 22) it is the opposite: CT26 responds well to anti-PD-1 therapy but MC38 does not respond...

Response:

Thanks for pointing out. Indeed, the Ref. 22 in the previous version did not show consistent anti-tumour efficacy of PD-1/PD-L1 antibodies against these two models as we and many other reports found. (please refer to the following publications:

Homet Moreno B, Zaretsky JM, Garcia-Diaz A, et al. Response to Programmed Cell Death-1 Blockade in a Murine Melanoma Syngeneic Model Requires Costimulation, CD4, and CD8 T Cells. *Cancer Immunol Res.* 2016;4(10):845-857. doi:10.1158/2326-6066.CIR-16-0060;

Kleinovink JW, Marijt KA, Schoonderwoerd MJA, van Hall T, Ossendorp F, Fransen MF. PD-L1 expression on malignant cells is no prerequisite for checkpoint therapy. *Oncoimmunology.* 2017;6(4):e1294299. doi:10.1080/2162402X.2017.1294299.

We apologize for not checking the results carefully in the reference. We assume the inconsistency could be due to the cell line maintenance in different labs. We confirmed our MC38 and CT26 cells via short tandem repeat (STR) profiling by authorized agents (data not shown) and had repetitive results in different experiments showing that CT26 is a moderate-responsive model while MC38 is a good-responsive model upon PD-1/PD-L1 therapy. Therefore, we relied on in-house results and kept this statement mentioned above in the current version. The citations were corrected

accordingly.

1.15-In line 210 it is stated that “In consistent with IBI319’s efficacy, both the PD-1 and CD137 were up-regulated on MC-38 tumor infiltrated T cells measured by FACS (Fig. S3C).” However I do not see a clear upregulation of CD137 in TILs compared to PBLs...Please modulate this statement or show data supporting this. Also in this figure the last two lines correspond to figure S4 legend...(they are duplic

Response:

Thanks for pointing out. We measured and plotted the percentage of CD137 and PD-1 expressing T cells in peripheral blood (Blood) and tumour (TIL) 11 days and 28 days after MC38 tumour implantation (current supplementary Fig. 3f). Compare to blood, CD137 up-regulated but with much less extent than PD-1 on TILs 11 days after inoculation. However, 28 days after tumour inoculation, CD137 positive cells increased and expressed on most TILs. Therefore, CD137 up-regulation on the TILs in MC38 tumour model increased with time. We modified the result description lines 241-244 of the current main body: “...more MC38 tumour-derived TILs expressed CD137 at 28 days after tumour inoculation than at 11 days after tumour inoculation (Supplementary Fig. 3f), suggesting that CD137 on TILs was upregulated with tumour progression.”

Reviewer #2 (Remarks to the Author):

In this manuscript entitled “PD-1 dependent CD137 co-stimulation as a novel strategy to enhance efficacy while avoiding off-tumor toxicity”, Qiao et al. designed and generated a CD137/PD-1 bispecific antibody (IBI319) which has higher affinity to bind to PD-1 and lower affinity to bind to CD137 to induce restrictive activation of CD137 in PD-1 rich microenvironment. Later, the authors conducted in-vitro cell based experiment and showed that the CD137 activation is dependent on the presence of PD-1. In addition, in vivo mouse model experiments were carried out to characterize and investigate IBI319’s activity and pharmacokinetics. The mouse model data showed that IBI319 has very promising antitumor activity with limited hepatotoxicity indicated by CD45 and F4/80 markers. Furthermore, preclinical study of pharmacokinetic and safety profile of IBI319 were also conducted in cynomolgus monkeys. Overall the study is well designed and conducted. However, some issues need to be addressed:

2.1-This is an important area. Would put the recent publication <https://pubmed.ncbi.nlm.nih.gov/33310772/> emphasizing irAE are important to avoid. Add it is important to get efficacy without irAE and how this molecule plays a role. Also refer to the recent publication <https://jitc.bmj.com/content/8/2/e001650> why 41BB is important for this pathway.

Response:

Thanks for pointing out. We agree that the two major rationales of this study shall be further emphasized. We re-emphasized the points and cited the relevant publications mentioned by the reviewer in the introduction part of the current version in lines: 43-50 (“A CD137 agonist further enhanced the anti-PD-1 antibody-mediated reinvigoration of exhausted CD8+ TILs from both primary sites and metastatic sites, indicating the rationale for targeting CD137 in combination with checkpoint blockade. However, the clinical trials evaluating two CD137-specific monoclonal antibodies (mAbs) were halted due to either intolerable hepatotoxicity (urelumab, BMS) or low efficacy (utomilumab, Pfizer). With the increasing number of clinical studies performed to evaluate

immunotherapeutic agents, it is important to avoid potential immune-related adverse events (irAEs) that could be life-threatening.”).

2.2-IBI319 was designed to possess relatively low affinity to bind to CD137 and therefore favor strong binding to PD-1. Fig 1C showed that IBI319 could compete and completely block the binding of CD137 ligand (CD137L) to CD137. In this Octet RED96e system binding kinetics and affinity assay, what’s the binding affinity of natural CD137L to CD137? Does IBI319 exhibit higher affinity than natural CD137L?

Response:

It’s a good point to compare the binding affinities of CD137 antibody with the natural ligand CD137L. We measured the affinity of IBI319 with two methods: SPR (Biacore T200, GE Healthcare, new data) and BLI (Octet RED96e, Fortebio), resulting K_D s of 394 and 719 nM, respectively. We have not measured the CD137 binding to CD137L in-house with the methods above, yet it’s been reported in (S. M. Chin, et al., Nature Communications 2018 (9); DOI: 10.1038/s41467-018-07136-7) that monovalent CD137L binds to CD37 with the K_D at 680 nM using SPR, which is comparable to IBI319 measured in-house. We referred this publication in the current Fig 1b and explained in the main body lines 89-98. In the current version we showed the SPR affinities in the main Fig. 1b and moved the BLI affinities into supplementary Fig. 1a.

2.3-Figure 3D showed the ratio of different cell clusters in each treatment groups, was there any data showing the absolute cell clusters numbers/amounts to compare each cell clusters between treatment groups?

Response:

Thanks for pointing out. We have incorporated the absolute cell number of each cluster in the sauce data_ Fig3c (sauce data was uploaded with the submission). Of note, the absolute cell number was relatively low for these samples in single cell analysis, as tumour cells occupied the majority of the live cell suspension sample. And the cell number per sample was limited for the single cell sequencing method.

2.4-Currently a few PD-L1/CD137 bispecific antibodies are under clinical development. What’s the rationale to choose PD-1 over PD-L1/L2 to design IBI319 in this study? Have the authors compared IBI319 with other PD-L1/CD137 bispecific antibodies, whether the other PD-L1/CD137 bispecific antibodies’ CD137 activity depend on PD-L1/L2?

Response:

Indeed, PD-L1/CD137 and PD-1/CD137 bispecific antibodies share the rationale of blocking PD1/PDL1 and agonizing CD137 pathways. We chose PD-1 over PD-L1/L2 to design IBI319 due to:

i) Targeting PD-1 blocks its binding to PD-L1 as well as PD-L2 while targeting PD-L1 only blocks PD-1/PD-L1 interaction. PD-L2 is a second ligand for PD-1 and is widely expressed by various tumours (Latchman, Y., et al. PD-L2 is a second ligand for PD-1 and inhibits T cell activation. Nat Immunol (2001). doi:10.1038/85330; Yearley JH, et al. PD-L2 Expression in Human Tumours: Relevance to Anti-PD-1 Therapy in Cancer. Clin Cancer Res (2017). doi: 10.1158/1078-0432.CCR-16-1761; Duan J, et al. Use of immunotherapy with programmed cell death 1 vs programmed cell death ligand 1

inhibitors in patients with cancer: a systematic review and meta-analysis. JAMA Oncol (2020) doi:10.1001/jamaoncol.2019.5367);

ii) PD-L1 is expressed on not only tumour cells, but also on normal tissue and immune cells such as DC and macrophages. PD-L1/CD137 bispecific antibody may mediate an extended immune synapse formation between e.g. T and DC cells, leading to an accelerated T cell exhaustion. The PD-L1 on normal tissues could be a potential risk for unexpected side effects of PD-L1/CD137 bispecific antibodies.

iii) We'd like to directly provide 4-1BB co-stimulation on PD-1+ T cells, with the aim to extend the durability of T cell reinvigoration and hopefully memory formation.

However, PD-L1/CD137 bispecific antibody may have advantage in specific targeting due to the broad PD-L1 expression on tumour cells. The differential efficacy and safety profiles between PD-L1/CD137 and PD-1/CD137 remain to be revealed and validated in clinical studies. Please see the relevant revision in the discussion part of the current version lines 353-366.

Regarding the comparison with other PD-L1/CD137 bispecific antibodies, we did not generate PD-L1/CD137 bi-specific antibodies in house, but investigated publications and patents from known PD-L1/CD137 bi-specific antibodies at clinical stage. According to publications and patents from a few molecules (GEN1046 (patent WO2019/025545), INBRX-105 (patent WO2017123650A2 mcla-145 (patent WO2018/056821), FS222 (Matthew A. Lakins, et al., Clin Cancer Res August 1 2020 (26) (15) 4154-4167; DOI: 10.1158/1078-0432.CCR-19-2958)), the efficacy of PD-L1/CD137 bi-specific antibodies all highly relied on PD-L1 expression.

Besides, with the purpose of comparing CD137 antibodies with different binding epitopes and affinities, we generated two other PD-1/CD137 bi-specific antibodies in-house using anti-CD137 sequences from urelumab and uomilumab. Indeed, all tested anti-CD137 arms, regardless of affinity, showed activity dependent on the presence of PD-1 arm. (supplementary Fig. 2g and lines 193-199 of the main text). These results could be due to the nature of CD137 relying on receptor trimerization.

2.5 What's the current status of IBI319? The preclinical data seems very promising, is it under clinical development? It would nice to see path forward.

Response:

IBI319 is currently under the dose escalation stage of Phase I clinical study (no. CTR20210027) in China. Thanks for the comment and we are also looking forward to the clinical readout in future.

Reviewer #3 (Remarks to the Author):

This is an interesting publication introducing a novel 1+1 PD1/CD137 bispecific antibody with an interesting design: High PD-1 affinity and low CD137 affinity. This is to my knowledge the first report on a PD-1 targeted CD137 agonist and the 1st BsAb targeting PD-1 and CD137 in clinical trials working through PD-1 inhibition and subsequent cis/trans 4-1BB activation. The studies are generally well performed and insightful, however, some experiments lack suitable controls.

3.1- From the experiments one remaining question is whether the dose response of PD-1 inhibition on the one hand and CD137 agonism on the other hand match to each, or in other words, is the dose required for PD-1 blockade also the optimal dose to agonize CD137. This could only be shown comparing to a combination of maximal PD-1 inhibition and maximal CD137 agonism, however, for that purpose e.g. the use of Fc dependent CD137 antibodies is required.

Response:

This is a very important point in general when combining two antibodies into a bi-specific antibody as the molar ratio will be one to one and not adjustable. In the specific design of IBI319, the efficacy of the anti-CD137 arm shall meet the criteria that it is dependent on the anti-PD-1 arm, is not a super agonist which induces liver toxicity and is optimal to agonize CD137. We showed data supporting the first two points as in Fig. 2b, d and Fig. 3f, respectively. For the third point which is to prove the efficacy of CD137 arm in vivo, following the suggestion, we constructed anti-CD137 mAbs with Fc functions (IgG1-Fc-wt). We kept human IgG1 and not mouse IgG1/IgG2a to have the molecules comparable to IBI319. For an even better comparison, the Fc active anti-CD137 mAb was monovalent with another arm binding to hen egg white lysozyme (Hel) - CD137/Hel-Fc. The anti-Hel arm didn't recognize any human nor mouse antigens and was used to keep the IgG structure. Consistent with previous reports using weak CD137 binders, CD137/Hel-Fc became more efficacious than CD137/Hel-Fc-null (TGI: 63.1% and 30.2%, respectively) at 10 mg/kg (supplementary Fig. 3g). The CD137/Hel-Fc showed a tumour inhibition index (TGI) of 48% at 3 mg/kg which is similar as PD-1/Hel at the same dose (32%), suggesting a relatively balanced and optimal combination. At 3 mg/kg, IBI319 showed stronger efficacy than PD-1/Hel or CD137/Hel-Fc alone, indicating the function of anti-CD137 mAb and the synergy of the two arms (Fig. 3e). We also tested CD137 binders with higher affinities using sequences from urelumab (Ure) and utomilumab (Uto). In *in vitro* T cell co-stimulation assay, IBI319 showed a best agonist effect comparing to PD-1/Ure and PD-1/Uto (supplementary Fig. 2h), suggesting that an optimal CD137 binder in a bispecific molecule requires a good combination of various factors, including affinity, the binding epitope and probably the molecular format.

Therefore, these results indicated i) both arms of IBI319 were functional in vivo; ii) the CD137 arm significantly enhanced the efficacy of PD-1 arm in IBI319, iii) The CD137 and PD-1 binders were relatively optimal in efficacy against MC38 tumour and iv) The Fc function was required for the anti-CD137 mAb as mono-agent, but not a must for the combination of anti-CD137 mAb and anti-PD-1 (Fig. 3a, b, e). We are aware that due to the different expression panels of PD-1 on TILs and Fc receptors on NK and myeloid cells, the efficacy of PD-1-dependent and Fc-dependent CD137 agonists could differ in different in vivo tumour models and this remain to be explored in future studies.

Relevant text in main body: lines 252-267 and 193-204.

3.2 The authors should 1) try to use more physiologically relevant T cell models in vitro/ex vivo and 2) in the in vivo studies rather compare to an Fc active CD137 antibody that mediates stronger liver tox but also stronger efficacy. The concern with the in vivo studies as they are is that either PD-1 + CD137 is not tested (CT26) (and thus it is not shown how good agonizing both pathways can be) or the superiority of combining PD-1 and CD137 over the single agents is not shown (MC38).

Response:

- 1) We added T cell memory recall assay as a more physiologically relevant in vitro experiment: Since antigen-specific T cells contribute to the recognition of tumour antigens and may upregulate PD-1 and CD137 in the tumour site, these cells could serve as a more physiologically relevant model. Therefore, we evaluated the co-stimulatory function of IBI319 in a cytomegalovirus (CMV) memory recall assay using PBMCs from three donors previously infected with CMV, as T cells specific for CMV antigens comprise a fraction of the memory T cell pool. The CMV peptide pool could induce IFN- γ production, and IBI319 further enhanced this effect better than α PD-1 or the combination of α PD-1 and α CD137, indicating the contributions of both arms of IBI319 (Fig. 2f and supplementary Fig. 2e). Relevant text in main body: lines 182-190.
- 2) For in vivo study, we have done the following revision works:
 - a) (old data incorporation) We added one result using CT-26 in which the PD-1 + CD137 combination group was included (supplementary Fig. 3c). The PD-1 + CD137 at equivalent dose showed a synergistic efficacy similar as MC38 model (Fig. 3b).
 - b) (new experiment) In order to confirm the efficacy of antiCD137 arm and the superiority of combining PD-1 and CD137 mAbs over the single agents, following the suggestion, we have constructed mono-valent anti-CD137 mAbs with and without Fc function (CD137/HeI-Fc, CD137/HeI-Fc-null). The experiments and discussions are described in the response to comment no. 3.1.

3.3 Similarly, in the MLR assay the BsAb is not superior to PD-1 monotherapy, but just to the combination, which weakens the findings reported.

Response:

Indeed, IBI319 induced stronger IFN γ production than PD-1 monotherapy in a statistically not significant manner and only in high concentrations of the MLR assay. This could be due to the fact that in an MLR assay PD-1 and CD137 were not optimally expressed by the T cells. Following the suggestion, we have confirmed the data in a more physiologically relevant assay, the CMV memory recall assay as mentioned above in the response to comment no. 3.2 (-1)) to show the agonistic function of CD137 in TCR specific T cells (Fig. 2f and supplementary Fig. 2e).

3.4- Generally, if these point can be addressed having a bispecific antibody that induces CD137 agonism through PD-1 binding and independent of Fc γ R is of course advantageous and a very good approach. Particularly, as conventional 1st gen CD137 agonists have been reported to be associated with liver tox I am feeling that the current data set still lacks some elements to unequivocally demonstrate the CD137 contribution in the BsAb to checkpoint inhibition. In general, but maybe not as a requirement for this paper, it would also be nice to show superiority or at least equivalency of the BsAb over the combination of a PD-1 antibody with 2nd gen 4-1BB agonists that lack liver tox or e.g. PD-L1 targeted CD137 agonists. Ultimately, given the tox of conventional CD137 agonists,

showing that the BsAb is equally efficacious as the combination would be sufficient, e.g. for the BsAb not necessarily better efficacy needs to be shown.

Response:

These are very important points. To avoid liver toxicity of the 1st gen CD137 super-agonist such as urelumab, and to have superiority in efficacy over the 2nd gen CD137 agonist and checkpoint inhibitor alone were indeed the two major goals in developing this bsAb. In other words, we aim to achieve the following two criteria: 1) bsAb is superior in efficacy than either of the single treatment and 2) bsAb is equally efficacious as the combination but avoid liver toxicity.

For 1): we have added new in vitro (CMV memory recall assay, (Fig. 2f and supplementary Fig. 2e) and in vivo experiments (CD137/Hel-Fc, Fig. 3e) further supporting the original conclusions in the manuscript. Please see details in response to comment no.3.2-1) and 3.1, respectively.

For 2): in terms of the 2nd generation CD137 agonists, we consider our CD137 mAb is one of them as it blocks the natural ligand binding epitope and is with relatively low affinity like CD137L (Fig.1b, c). However, from our result, although there was neither clear efficacy nor severe toxicity of the α CD137 mAb (Fc-null) alone, the liver toxicity emerged clearly when its combined with α PD-1 mAb. We assume this could be due to a systematic, sequential effect of CD137 upregulation on various immune cells upon PD-1 blocking on T cells. Therefore, although we didn't combine PD-1 blockade with other 2nd gen CD137 agonists from other groups (e.g. utomilumab), the risk of triggering toxicity by doing so still exists.

With the results in this report, we hope to convince the reviewers that the CD137 arm of IBI319 is functional, and IBI319 has equivalent efficacy but non-detectable liver toxicity comparing to the combination.

3.5 - Another important question is whether the affinity chosen for the CD137 arm is optimal, or whether better CD137 agonism could be achieved using a slightly higher affinity antibody e.g. 50 or 100 nM would be advantageous given the relatively low levels of PD-1 available for crosslinking

Response:

In order to further test if the CD137 arm of IBI319 is optimal, we constructed PD-1/CD137 bispecific antibodies using anti-CD137 sequences from urelumab (PD-1/Ure) and utomilumab (PD-1/Uto). PD-1/Ure and PD-1/Uto bound to CD137 with K_{DS} at 19.2 nM and 49.3 nM, respectively. In the Jurkat-CD137-luc reporter assay, IBI319 was equivalently efficacious as PD-1/Uto. PD-1/Ure showed the highest activity among all tested bispecific molecules (Supplementary Fi. 2g). Interestingly, IBI319 showed a better primary T cell co-stimulation than the other 2 bispecific molecules (Supplementary Fig. 2h). The inconsistency in the link between efficacy and affinity remains to be resolved in the future but could be due to differences in the binding epitope, which would influence the crosslinking of two targets/cells. Therefore, the optimal CD137-specific sequence in a bispecific molecule requires a good combination of various factors, including affinity, the binding epitope and probably the molecular format. Relevant text in main body: lines 252-267.

3.6- Last but not least, the text should be more thoroughly checked by a native English speaker, I included some examples below.

Response:

We did language improvement through professional agency for the revised version.

General points

3.7-Include details on design and production of the BsAb, how is correct light chain association achieved? In vitro assembly of the BsAb?

Response:

Mutations introduced to heavy and light chains and IgG1_CH3 domain to generate IBI319 in order to ensure heterodimerization and correct heavy / light chain pairing were previously reported in (Leaver-Fay et al., 2016 Structure 24, 641-51, Lewis et al., 2014 Nat Biotechnol 32, 191-8) and described in the Method section “Antibodies, proteins, cells and mice” of the current main body. The mutations are listed as the table below.

Sequential (Kabat/EU) numbering of IBI319	Mutations (Kabat numbering)	Purpose
Anti-CD137 HC	Q39Y, 105R*, S131C, K218D, C220G, E356G, E357D, S364Q, Y407A	Heteromab assembly
Anti-CD137 LC	Q38R, A1D, K42D D122K	Heteromab assembly
Anti-CD137 HC	L234A, L235A, N297Q	FcγR Effector assembly
Anti-PD-1 HC	Q39K, K62E, H168A, F170G, Y349S, T366M, K370Y, K409V	Heteromab assembly
Anti-PD-1 LC	D1R, Q38D, L135Y, S176W	Heteromab assembly
Anti-PD-1 HC	L234A, L235A, N297Q	FcγR Effector ablation

*Typically, amino acid 105 (Kabat numbering) is mutated to an R for heteroMAb assembly. However, the anti-CD137 HC already contains an R at this position.

3.8-What is the expression level of PD-1 on CHO-S-PD-1 and how does it compare to natural PD-1 expression, this may be important to show the relevance as in general PD-1 expression is low and the question is how much 4-1BB agonism can be achieved via crosslinking through PD-1.

Response:

We analyzed the expression levels of PD-1 and CD137 in the following experiment settings:

- 1) The absolute cell surface expression level of PD-1 on CHO-S-PD-1 and CD137 on Jurkat-CD137 and Jurkat-CD137-NFκB-Luc2P-PD-1 were measured using Qifikit and compared with PBMC-derived T cells after CD3/CD28 beads stimulation (Supplementary Fig. 2a, lines 148-156). PBMC-derived T cells after stimulation expressed comparable CD137 as CD137 cell lines, comparable PD-1 as Jurkat-CD137-NFκB-Luc2P-PD-1, and less PD-1 as CHO-S-PD-1 cells. The relative expression comparison via FACS was shown in Supplementary Fig. 1d-right from 3 different PBMC donors (lines 103-104).
- 2) The relative cell surface expression level of PD-1 and CD137 on PBMC-derived T cells after sub-optimal CD3/CD28 stimulation (the setting for T cell co-stimulation assay, supplementary Fig. 2d-bottom) was analyzed using FACS. Less than 50% of CD4+ and CD8+ T cells expressed PD-1, which possibly explained why only at the presence of PD-1 expressing CHO cells, could IBI319 further enhance T cell activation (Fig.2e, Supplementary Fig. 2d-bottom, lines 179-180).

- 3) The CD137 and PD-1 expression level on T cells in peripheral blood (blood T) and tumour (TIL) were analyzed 11 and 28 days after MC38 tumour inoculation (Supplementary Fig. 3f) in C57BL/6-hPD-1/hCD137 mice. PD-1 expression was clearly upregulated on TIL (58-74% on CD8+T cells and 25-35% in CD4+ T cells) as compared with blood T. CD137 expression was also up-regulated but with much less extent than PD-1 on TILs 11 days after tumour inoculation. However, 28 days after tumour inoculation, CD137 further increased and expressed on most TILs (Supplementary Fig. 3f). This data suggested that the PD-1 high expression, especially on CD8+T cells could mediate the crosslinking of IBI319 (lines 241-244).

3.9 - Figure 1E following: Considering, that you see hardly binding to 4-1BB with the α CD137 (red) antibody it is reasonable to assume that you have hardly any 4-1BB expressed on this cells (of course to claim this a high affine α CD137 control is missing here).

Response:

As mentioned above in the response to comment no. 3.7, PBMC-derived T cells which were used in the binding assay in Fig. 1e, expressed a certain level of CD137. However indeed, the binding of anti-CD137 mAb to these T cells was very weak and varied among PBMC donors. In Jurkat-CD137 cell line which expresses more homogeneous and higher level CD137, the binding of α CD137 mAb was detectable but also weak (Fig. 1f). It was previously reported that the trimerized CD137L-CD137 complex ensure the recruitments of downstream signal mediators which could further stabilize the binding (S. M. Chin, et al., Nature Communications 2018 (9); DOI: 10.1038/s41467-018-07136-7, A. Bitra, et al., J Biol Chem. 2018 (26); DOI: 10.1074/jbc.RA118.003176). Therefore, we assume the CD137mAb low binding to primary T cells is due to a) low affinity, b) relatively low CD137 expression level on T cells and c) possibly the mAb without crosslinking mechanism lacks the capability to stabilize CD137. Relevant text in main body: lines 103-113, 114-125.

3.10 - Furthermore, seeing than the strong MFI increase (with a increase in EC50) of the bispecific IBI319 (green) compared to α PD-1 (blue) one could also assume that the difference is due to the bivalent to monovalent binding to PD-1 and not additive effect of bispecific binding. E.g. a high affine monovalent binder will double the MFI compared to a high affine bivalent Ab (as the double amount of Fc (Ab) will be clustered on the cell surface).

Response:

We agree that the difference of IBI319 comparing to α CD137 mAb in binding to primary T cells could be mainly due to the bivalent to monovalent binding to PD-1, and the IBI319 binding is mainly contributed by the anti-PD-1 arm. Therefore, in order to dissect the cell based binding to CD137, the Jurkat-CD137 cell line was used in Fig. 1f, and the results revealed that IBI319 bound to CD137 in a PD-1 dependent manner. Of note, the α CD137 mAb showed a slightly higher affinity to Jurkat-CD137 cells, presumably due to the higher and more homogenous CD137 expression on Jurkat-CD137 cells than the primary T cells. Relevant text in main body: lines 114-125.

3.11 Figure 2A, where does the weak activity of PD-1 antibodies come from? They should not affect NFkB signaling per se, is this just background?

Response:

Yes the weak activity of anti-PD-1 mAb in Jurkat-NFKB-Luc assay is considered as nonspecific background (added to the current version line 144).

3.12- This cell line monitor only NFkB and therefore 4-1BB activity. But not the influence of PD-1 blockade. This could be made clearer (like Fold induction of NFkB activation).

Response:

Thanks for pointing out. We further clarified the mechanism of this experiment in the result section of the main body as "To specifically evaluate antibodies triggering CD137 signalling, the Jurkat-CD137 cells was co-cultured with antibodies with or without CHO-S-PD1 cells. Since the luciferase activity in Jurkat-CD137 cells was regulated by NFkB, its induction reflected only CD137 function and not the influence of PD-1 blockade". (added to the current version lines 134-138)

3.13 -Does Jurkat-CD137-NFkB-Luc2P-PD1 display the same CD137 expression as Jurkat-CD137-NFkB-luc2? Does Jurkat-CD137-NFkB-Luc2P-PD1 display the same PD1 expression as CHO-S-PD1? Clarify:

- 1) Jurkat-CD137-NFkB-Luc2P and the Jurkat-CD137-NFkB-Luc2P-PD1 display the same activation potential (like stimulation with TNFa)
- 2) Jurkat-CD137-NFkB-Luc2P and the Jurkat-CD137-NFkB-Luc2P-PD1 display the same expression of CD137
- 3) Jurkat-CD137-NFkB-Luc2P-PD1 and CHO-S-PD1 display the same expression of PD1?

Response:

Please see the clarifications as below:

Regarding 1), Jurkat-CD137-NFkB-Luc2P-PD1 was generated on purpose that the stable clone had a similar CD137 expression level as Jurkat-CD137-NFkB-Luc. An TNFa stimulation assay revealed a similar activation potential of these two cell lines (Supplementary Fig. 2b and lines 156-157 of the main body).

Regarding 2): QIFIKIT quantification revealed that the CD137 expression levels on Jurkat-CD137-NFkB-Luc (Jurkat-CD137 for short) and Jurkat-CD137-NFkB-Luc2P-PD1 were similar (Supplementary Fig. 2a and lines 148-153 of the main body).

Regarding 3): Using QIFIKIT quantification, CHO-S-PD1 expressed higher level of PD1 than Jurkat-CD137-NFkB-Luc2P-PD1 (around 30 times higher), indicating a potentially stronger capability of mediating IBI319 crosslinking via *trans*- cell bindings. (Supplementary Fig. 2b. and lines 156-157 of the main body).

3.14-Fig 2C: The picture is a little bit misleading as it implies that you did not have seeded the same amount of Jurkat-CD137-NFkB-Luc2P (4 light dots) and the Jurkat-CD137-NFkB-Luc2P-PD1 (7 red dots). However for a fair comparison the cell number of the reporter cell lines should be the same. Please clarify

Response: Thanks for pointing out. Indeed, we seeded the same amount of reporter cells in this experiment. We corrected the drawing in the current Fig. 2c-left.

3.15- Fig 2D: 1) The CD137 antibody used here is the Fc-dead version, if yes, it is a superagonist that works without FcγR binding? 2) It seems that the MLR affect is driven by PD-1 inhibition, does the combination with CD137 add anything significant This may be a consequence of CD137 being Fc dead? 3) Actually, in all cases the BsAb is not significantly different from PD-1 alone ->

conclusions that it is superior to the combination need to be adapted, there is sometimes significance compared to the combination, but not to single agent PD-1? So, in Fig 2D the BsAb appears not better than PD-1 alone, it is only if CHO-PD-1 is added, but not in a classical MLR. It would be better to show this via PD-1 expression on T cells superior MLR activity can be seen

Response:

- 1) Yes, the α CD137 mAb is an Fc-dead version. In Fig. 2d the α CD137 mAb did not show agonist activity in MLR. We have changed the color code of this figure to make the expression clearer.
- 2) Yes, PD-1 inhibition drove the majority of MLR effect in Fig. 2d. Fc dead could be the main reason of α CD137 mAb lacking efficacy in vitro. From the new in vivo data, the Fc active version of anti-CD137 mAb was efficacious (Fig. 3e and supplementary Fig. 3g, please refer to the response to comment no. 3.1).
- 3) In the MLR assay, although not statistically significant, we did see a stronger effect from IBI319 than α PD-1 mAb alone, especially at high concentrations. Knowing that IBI319 is monovalent for anti-PD-1 thus only has half of the anti-PD-1 molecules as bivalent α PD-1 mAb, this data indicated the contribution from the anti-CD137 arm. In addition, as mentioned above in the response to comment no. 3.2-1), we added a T cell memory recall assay to further evaluate the superiority of IBI319 over α PD-1 mAb.

3.16 In vivo studies: The CD137 antibody used here has no Fc function, correct?

If an Fc dead CD137 antibody is used where does the liver tox come from, also an Fc dead one should not be efficacious?

Does the antibody exhibit super-agonism?

In general, from the data it appears the CD137 antibody is largely inactive regarding 4-1BB agonism?

If this is true it may not be the ideal comparator and personally a Fc active CD137 antibody may be the better comparator to show that equal or better efficacy can be achieved without liver tox with the BsAb. Please clarify in the text the lack of Fc function.

Response:

we clarify that the α CD137 mAb used in the first submitted version has neither Fc function, nor is super agonist. Following the suggestion, in order to confirm the efficacy of antiCD137 arm and the superiority of combining PD-1 and CD137 mAbs over the single agents, we have constructed CD137 mAbs with Fc function. The results and interpretations were described in the response to comment no. 3.1.

Regarding the toxicity induced by Fc-null CD137 antibody, the clear toxicity only present in the animals treated by the combination (α CD137 + α PD-1 mAbs). We assume this could be due to a systemic, sequential effect of CD137 upregulation on various immune cells upon PD-1 blocking on T cells. We didn't observe strong liver toxicity from the mice treated by the anti-CD137 with Fc function (CD137/Hel-Fc) at 3 mg/kg (Supplementary Fig. 5e), presumably because of low affinity of this CD137 binder as well as low antibody dose (relevant text lines 273-276).

3.17-Fig 3A. The combination as a comparator is lacking; any explanation why it prevents relapse, outgrowth at later times, but early on is not superior to PD-1

Response:

We incorporated a separate experiment of CT26 model in Supplementary Fig. 3c, in which the

combination group was included (relevant text lines 229-230). Regarding IBI319 preventing relapse in Fig. 3a, we speculate two explanations: a) CD137 expression in TILs could increase with time (shown in MC38 tumour in Supplementary Fig. 3f) (relevant text lines 241-244); b) CD137 agonist may preferentially enhance memory T cell function, which could then prevent tumour growth for a longer time after treatment termination. This assumption was supported by the CMV T cell memory recall assay (Fig. 2f and supplementary Fig. 2e) (Relevant text lines 182-190).

3.18- Fig 3B: the comparison to PD-1 monotherapy alone in the MC38 model is missing, it could be that the efficacy comes just from PD-1 inhibition alone, and not from PD1- + CD137 combination to which it is compared; is the BsAb really better than PD-1 in MC38? In the supplement PD-1 monotherapy data is shown but only for 3 mg/kg,, not for the combination dose

Response:

In Fig. 3B (current Fig. 3b), the superiority of IBI319 to PD-1 inhibition alone was indeed not very clear. One reason could be that the α PD-1 mAb is bivalent thus the molar dose of anti-PD-1 is two folds of IBI319, in which the anti-PD-1 arm is monovalent. To better address this question, i.e., whether or not the anti-CD137 arm of IBI319 is functional, we introduced monovalent CD137/Hel-Fc with Fc function and PD-1/Hel and compared their efficacy with IBI319. Please refer to the response to comment no. 3.1 and results in Fig. 3e.

3.19 Fig 3E: clarify in the text that CD137 mAb had no Fc function

Response: The CD137 mAb had no Fc function is clarified in the main body lines 101-102 (For control purpose, the Fc γ R binding function of α CD137 mAb is also silenced...) and in Fig. 1b.

3.20- Fig 4A. Explanation for drop of antibody levels around 240-360h; is this ADA related, explain.

Response: Yes, the drug concentration dropped and likely this was related to the occurrence of ADA. The blood sampling time for ADA detection was on D1, D8 (~192h), D15 (~360h), D29 (~696h) and D36 (~864h). The frequent occurrence of ADA was detected on D15 and persisted until D36, verified by the high ADA positive rate in all treated groups. The ADA titer ranges in 0.1, 1, 10 mg/kg groups at D15 were 1:640~1:10240, 1:640~1:10240, 1:320~1:2560, respectively. Relevant data was incorporated into Fig. S6A and explained in lines 295-299 of the main body.

3.21 Discussion: A few other molecules were reported to implement similar designs, PRS-343 (HER2/CD137, Pieris Pharmaceuticals) and two molecules from Roche (FAP/4-1BBL and CD19/4-1BBL) combine a tumor associated antigen (TAA) with CD137 agonist or 4-1BBL7,25, but their efficacy as a single agent might be limited since the TCR signal 1 to primarily activate T cell is missing. => discuss possibility to combine these agents with PD-1/PDL1 inhibition to re-activate the signal 1.

Response: We added relevant text in the discussions section lines 352-353: "Thus, for these molecules, combination therapy with PD-D/PD-L1 inhibitors or other agents, such as CD3 bispecific T cell engagers, could potentially activate signal 1."

3.22 Wording/English points

in general: replace "remain" by "retain" throughout the text

- we have replaced accordingly as indicated in the text

The following sentences would need re-writing as there are errors or they are difficult to understand Nevertheless, to simultaneously reduce the off-target toxicity while RETAINing anti-tumor efficacy are the remaining challenges to advance CD137 agonists in clinical applications, and to solve which the Fc function, affinity and binding epitope shall be all considered in a reasonable manner for the desired new molecule. (2nd sentence cannot be understood)

- New version: Nevertheless, reducing off-target toxicity while retaining antitumour efficacy is a continuing challenge in advancing CD137 agonists into clinical applications, and overcoming this issue will likely require consideration of the Fc function, affinity and binding epitope properties of the desired new molecule (lines 56-59).

to its ligands PD-L1/PD-L2 provides a negative feedback signal against TCR activation via protein tyrosine phosphatase SHP-2, the inhibition of which via anti-PD-1 /PD-L1 antibodies is mainly through signal (something is missing in the 2nd sentence, also please note the role of CD28 co-stimulation in order for CPI to work)

- New version: Since the binding of programmed cell death 1 (PD-1) to its ligands programmed death-ligand 1 and 2 (PD-L1/PD-L2) provides a negative feedback signal to counteract TCR activation via the protein tyrosine phosphatase SHP-2, PD-1/PD-L1 blockade mainly affects the outcome of signal 1 (lines 62-65).

Therefore, simultaneously blocking PD1/PD-L1 and activating CD137

- Corrected new version: simultaneously blocking PD1/PD-L1 while activating CD137 (lines 65-66)

arm is from Sintilimab (α PD-1), an approved PD-1 blockade??

- Corrected new version: arm is from Sintilimab (α PD-1), an approved PD-1 blocker,... (line 85).

indicating an overlapping binding epitope

- Corrected new version: indicating the antibody binding epitope overlapped with the binding site of the natural ligand (lines 95-96).

strong blocking OF PD-1 and appropriate agonizing OF CD137

- Corrected new version: strong blocking of PD-1 and appropriate agonism of CD137 (line 93).

and the induced PD-1/CD137 on PBMC is relatively low.

- Adjusted new version: We assumed that the low binding of the α CD137 mAb to primary T cells was due to its low affinity, **the lower CD137 expression level on T cells than on the cell line tested (Supplementary Fig. 1d)** and possibly the bivalent mAb without crosslinking lacking the capability to stabilize the binding with CD137...(lines 108-110)

Therefore, in addition to confirm IBI319 in remaining this synergy, we ...

- Corrected new version: Therefore, in addition to confirming IBI319 retains this synergy, we...(lines 214-215).

In consistent with IBI319's efficacy

- Corrected new version: Consistent with the efficacy of IBI319 (line 239)

Further, the bispecific format of IBI319 has shown it comparable efficacious as the combination of the two agents in in vivo setting.

- Adjusted new version: The simultaneous targeting of PD-1 and CD137 by the bispecific format of IBI319 showed antitumour efficacy *in vivo* (lines 327-328).

REVIEWERS' COMMENTS

Reviewer #1 (Remarks to the Author):

The authors have properly addressed all my comments and the manuscript has been considerably improved.

Just a small thing that needs to be corrected:

Lines 223-224. Please correct this sentence: "Of note, it was previously reported that murine PD-L1 (mPD-L1) binds to human PD-1 (hPD-1) with an affinity similar to that of hPD-L1 binding to hPD-L1". The last term should be hPD-1 instead of hPD-L1.

Reviewer #2 (Remarks to the Author):

PI see below

Reviewer #3 (Remarks to the Author):

The authors have addressed the comments raised with additional experiments and clarification. No further revisions are required from my side

REVIEWERS' COMMENTS

Reviewer #1 (Remarks to the Author):

The authors have properly addressed all my comments and the manuscript has been considerably improved.

Just a small thing that needs to be corrected:

Lines 223-224. Please correct this sentence: "Of note, it was previously reported that murine PD-L1 (mPD-L1) binds to human PD-1 (hPD-1) with an affinity similar to that of hPD-L1 binding to hPD-L1". The last term should be hPD-1 instead of hPD-L1.

Response: Thanks for pointing out, the term has been corrected in the final manuscript.

Reviewer #2 (Remarks to the Author):

PI see below

Reviewer #3 (Remarks to the Author):

The authors have addressed the comments raised with additional experiments and clarification. No further revisions are required from my side